# Pain modulation by your partner: An experimental investigation from a social-affective perspective

**Katrin Hillmer**, **Judith Kappesser**, **Christiane Hermann** *

Department of Clinical Psychology, Justus-Liebig-University Giessen, Giessen, Germany

* Christiane.Hermann@psychol.uni-giessen.de

**Data Availability Statement:** The data which were collected in this experiment are not freely and directly available because the original approval by the local ethics committee of the Faculty of Psychology and Sports Science, Justus-Liebig

## Abstract

### Background

Social context such as the relationship between a person experiencing pain and a caregiver has been shown to affect the experience of pain, yet, results are not consistent. Possibly, differential effects of interpersonal relationships are modulated by affective states expressed by social partners. Viewing partner pictures in experimental designs is not only associated with lowered perceived pain intensity, but also affects neural responses. However, the role of affective modulation is not clear. The present study aimed to systematically examine the pain modulating effects of stimuli varying in affect and social content including personal relevance using subjective report and psychophysiological measures of facial and autonomic activity.

### Methods

Twenty-nine women underwent a tonic heat pain paradigm with simultaneous picture viewing to investigate the influence of their partners' faces with a neutral facial expression compared to strangers' happy, angry and neutral facial expressions on pain intensity and accompanying psychophysiological parameters (facial activity: corrugator muscle activity, autonomic activity: skin conductance level, heart rate). In addition to perceived partner support and relationship characteristics, the contribution of the affective value (valence, arousal) of the partner faces to the observed pain modulation was examined.

### Results

Partner and happy faces reduced self-reported pain intensity and corrugator activity, the latter being lowest when viewing partner faces as compared to all other picture categories. As corrugator activity is indexing stimulus unpleasantness and a core feature of the facial pain expression, this physiological pattern matches well with the subjective ratings. Neutral objects, neutral and angry faces had no effect on pain self-report, although angry faces were rated as highly negative. Partner faces also led to increased skin conductance, being an index of motivational activation, and heart rate deceleration, possibly reflecting increased sensory intake. Partner-related pain modulation was primarily related to perceived arousal

University Giessen, Germany (#2010-0018) and the informed consent form signed by the participants did not include such direct access. However, the anonymized data will be made available to interested researchers upon request by the local ethics committee of the FB06 - Psychology and Sports Science. For this purpose its contact email (ethikkommission@fb06.uni-giessen.de) is given in manuscript.

**Funding:** The authors received no specific funding for this work.

**Competing interests:** The authors have declared that no competing interests exist.

of the partner's picture, i.e., the intensity of the activation of approach motivation, and pain-related catastrophizing.

## Discussion

Our results are partially consistent with emotional pain control models, especially regarding the modulatory influence of valence. Within the context of socially adaptive behavior, they particularly underline the social signal value of emotion and attachment figures. Clinically, our results imply that just looking at pictures of one's partner when undergoing acute painful procedures can have a robust hypoalgesic effect.

## Introduction

There has been much research on social influences on chronic pain, particularly of social support and solicitousness [1, 2]. The evidence, however, is inconsistent with respect to the direction of the pain-modulating effect. Some studies found solicitous partner and socially supportive others to enhance pain [3–8], corresponding well with assumptions of the operant model of (chronic) pain [9]. Yet, other studies found partners' punishing responses to lead to increased pain and social support to relieve pain [10, 11]. When reviewing experimental studies examining social pain modulation, Krahé and colleagues [1] found overall that positive interactions led to a decrease in pain-related responses, whereas negative, mixed or ambiguous interactions (including the mere presence of another person) led to an increased pain response.

One explanation for these differential effects of interpersonal relationships could be differences in the affective state induced by the social partner. This assumption is not only supported by studies manipulating affect such as threat by varying the social context [12–14], but also by research using social and affective pictures as stimuli. Three experimental studies showed that photos of partners as compared to strangers, acquaintances and/or objects reduced pain [15–17]. When interpreting their results, authors referred mainly to the pictures' affective value reasoning that partner pictures are perceived to be pleasant which in turn leads to a positive affect. Consistent with Motivational Priming Theory [MPT; 18], partner pictures attenuated the experienced pain. However, the pictures' valence and arousal, commonly measured in MPT studies, was not assessed in these studies. Furthermore, in these three studies either a picture of the partner while smiling was used [15] or the facial expressions were not specified [16, 17]. Therefore, the role of the induced affective state remains unclear. Not only have different facial expressions (i.e., sad, angry, neutral, happy) different pain modulating effects [10, 19–22], but the affective state induced by a picture of the partner independent of their facial expressions may vary depending on the quality of the relationship [15, 23, 24] or pain-related cognitions such as catastrophizing [25].

It would be particularly meaningful to demonstrate that social cues modulate pain not only at the level of subjective report, but also at the level of pain-related psychophysiological responses. Due to being a highly relevant biological stimulus, pain induces autonomic arousal, as indicated by higher skin conductance level (SCL) and heart rate (HR) [26, 27], i.e., a physiological response pattern which is characteristic for fight-flight behavior [21, 28]. Moreover, pain stimuli are associated with increased activity of the corrugator muscle, which is consistent both with findings of eyebrow squeezing being a core element of the facial expression of pain [29], and corrugator activity correlating with perceived aversiveness of stimuli [30]. Viewing

pleasant and unpleasant scenes as well as emotional facial expressions is associated with higher SCL as compared to neutral pictures [18, 30, 31]. Moreover, pleasant picture elicit less corrugator activity [30, 32], whereas unpleasant increase corrugator activity [20, 33–36]. HR responses to emotional pictures vary depending on valence, yet, are complex due to differences in time course [31–33]. Little is known about HR responses when participants are exposed to both pain and emotional stimuli. Interestingly, there is evidence that pain-related HR acceleration is attenuated while viewing pleasant pictures [37].

The main aims of the present study were (a) to examine the pain modulating effect of pictures varying in (I) their affective state (happy, neutral, angry) and (II) their social content (partner, stranger, object), and (b) to investigate this effect on the subjective (pain report) and physiological level (HR, SCL, corrugator EMG). An additional aim (c) was to examine the association of pain modulation with induced affect, catastrophizing, perceived partner support and relationship characteristics.

## Methods

### 1. Participants

Female participants were recruited by mass email at the local university. The study was advertised as 'investigating the effects of stress and being in a relationship on emotional processing of stimuli'. Interested students were screened for eligibility and exclusion criteria by a telephone interview. To be considered for inclusion, participants had to be (a) female (in order to control for gender effects), (b) older than 18 years, and (c) in a close relationship for at least 6 months. Criteria for exclusion were: (a) self-report of acute or a history of chronic pain, (b) current mental health problems, (c) organic diseases, (d) intake of psychotropic or analgesic drugs, and (e) the partner did not consent to being photographed. Seven of 36 tested participants had to be excluded from data analysis due to illness on the day of the experimental session ($N = 2$), missing data ($N = 1$), extremely low pain thresholds ($N = 2$), history of self-injurious behavior ($N = 1$), and a fire alarm during the experiment ($N = 1$) resulting in a total sample of $N = 29$ women. Data of the final sample can be requested by writing to ethikkommission@fb06.uni-giessen.de. All participating women were students and on average 23.34 years (SD = 2.89) old. The majority (69.9%) had been in their partnership for longer than two years (see Table 1).

For their participation the female students received either 15 € or course credit. The local Faculty 06 –Psychology and Sports Science ethics committee (LEK-FB06) approved the study protocol (# 2010–0018) and written informed consent was received from all participants.

### 2. Questionnaires

Questionnaires assessing catastrophizing, relationship satisfaction, and perceived partner support were administered online via *Unipark* (Questback GmbH, Oslo, Norway) prior to the experiment. Participants rated their current level of happiness in their relationship as a state measure on the day of the experiment.

**2.1 Catastrophizing.** *Pain Catastrophizing Scale (PCS [38]; German version [39]).* The PCS consists of 13 items measuring three different aspects of catastrophizing thoughts about pain. The three subscales are *Helplessness* (6 items, e.g.,: "*I feel, I can't stand it anymore*"), *Rumination* (4 items, e.g.,: "*I keep thinking about how much it hurts*"), and *Magnification* (3 items, e.g.,: "*I become afraid that the pain may get worse*"). Frequency of the thoughts are rated on a Likert scale from 0 (*never*) to 4 (*all the time*), with the total score ranging between 0 and 52. A total score $> 30$ is considered as clinically relevant [38]. Factor structure and psychometric properties for the German version are comparable to those for the English version. Internal

**Table 1. Sociodemographic characteristics, questionnaire data and valence and arousal ratings of the partner picture and their correlation with experimental measures of partner pain modulation (partner-PM).**

| | | *Mean/ Frequency* | *SD* | Experimental measures | | |
| --- | --- | --- | --- | --- | --- | --- |
| | | | | Partner-PM[1] | Partner valence | Partner arousal |
| | | | | *r (p)* | *r (p)* | *r (p)* |
| Partnership duration | 6–24 month | 9 | – | -.07 (.715) | -.03 (.884) | -.05 (.802) |
| | >24 month | 20 | – | | | |
| Relationship satisfaction (RAS) | | 4.29 | .69 | .20 (.295) | .08 (.694) | .35 (.060) |
| Current happiness with relationship rating (0–6) | | 4.38 | 1.29 | .34 (.072) | .07 (.730) | .08 (.690) |
| Pain catastrophizing (PCS) | | 18.72 | 9.20 | **.45 (.014)** | .29 (.133) | .29 (.131) |
| Helplessness | | 7.52 | 5.09 | **.46 (.012)** | .26 (.171) | **.40 (.032)** |
| Rumination | | 8.24 | 3.98 | .34 (.074) | .32 (.087) | .11 (.573) |
| Magnification | | 2.97 | 1.82 | .27 (.165) | .01 (.975) | .10 (.620) |
| Pain-related partner behavior (MPI-D) | | | | | | |
| Punishing response | | .47 | .77 | -.17 (.384) | -.25 (.192) | -.16 (.410) |
| Distracting response | | 2.49 | 1.23 | -.20 (.307) | -.01 (.943) | -.09 (.645) |
| Solicitous response | | 4.29 | .99 | .11 (.564) | .29 (.122) | **.43 (.019)** |
| Stress related partner support (DCI) | | | | | | |
| Supportive DC | | 19.90 | 3.03 | -.04 (.858) | -.16 (.405) | .11 (.580) |
| Negative DC (invers) | | 16.90 | 2.78 | .04 (.848) | -.03 (.868) | .30 (.110) |
| Delegated DC | | 7.34 | 2.02 | -.18 (.362) | .09 (.633) | .13 (.514) |
| Conjoined DC | | 17.72 | 3.85 | .06 (.759) | -.06 (.770) | .18 (.358) |
| Satisfaction with partner support | | 4.24 | .83 | .09 (.659) | -.16 (.412) | .27 (.160) |
| Effectiveness of partner support | | 4.00 | .80 | -.09 (.639) | -.13 (.495) | .13 (.496) |
| Ratings for partner picture during experiment | | 7.43 | 1.35 | **.40 (.034)** | | **.42 (.022)** |
| Valence (0–9) | | | | | | |
| Arousal (0–9) | | 4.28 | 2.37 | **.51 (.004)** | | |

Significant correlations are highlighted in bold. Abbreviations: DC(I) = Dyadic coping (inventory); MPI-D = Multidimensional Pain Inventory–German version; PCS = Pain catastrophizing scale; RAS = Relationship Assessment Scale.

[1] Partner-PM refers to the difference in pain intensity when viewing neutral faces versus partner pictures (Δ neutral–partner) at t3, i.e., partner-PM > 0 indicates a pain relieving effect of partners' faces.

consistency for the subscales of the German version are good (Cronbach's α = .67 for magnification, Cronbach's α = .88 - .92 for the other scales and the sum score [39]).

**2.2 Relationship satisfaction.** *Relationship Assessment Scale (RAS [40]; German version [41]).* The RAS is a single factor scale measuring the generic satisfaction in close relationships and marriages. It consists of seven items (e.g., *"How well does your partner meet your needs?"*, *"In general, how satisfied are you with your relationship?")* that have to be rated on a 5-point Likert scale from 1 (*low satisfaction*) to 5 (*high satisfaction*). The final RAS score is averaged across the seven items. In a German student sample, internal consistency (Cronbach's α = .82), retest reliability (men: $r_{tt.}$ = .78, women: $r_{tt.}$ = .86) and validity of the RAS were good [41].

*Current happiness.* Participating women rated how happy they currently felt in their relationship (*"If you think right now about it, how happy or unhappy are you with your partnership?"*) on a 7-point Likert scale ranging from 0 (*extremely unhappy*) to 6 (*perfect*). This item was taken from the German version [42] of the *Dyadic Adjustment Scale* (DAS; [43]), a well-established measure for assessing the quality of partnerships [42, 43].

**2.3 Perceived partner support.** *Stress-related partner support (Dyadic Coping Inventory: DCI [44]).* The women's view of their partners' social support in stressful situations was

assessed using the DCI. The DCI consists of four subscales: (a) Supportive dyadic coping (DC; five items, e.g., *"He gives me the feeling that he understands me, and that he is interested in my stress."*), (b) Negative DC (four items, e.g., *"He accuses me that I cannot handle stress good enough."*), (c) Delegated DC (two items, e.g., *"He takes over tasks and activities that I normally do, in order to relieve me."*), and (d) Conjoined coping (five items, e.g., *"We try to deal with the problem together and look for concrete solutions.")*. Two additional items assess the satisfaction with and effectiveness of the partner's and the conjoined dyadic stress coping, summarized as '*Evaluation of partner support*'. All items are rated on a 5-point Likert scale ranging from 1 (*very seldom*) to 5 (*very often*). The subscales of the questionnaire have good internal consistencies (Cronbach's α = .71 - .92). The retest-reliability varies between .53 and .80. validity was shown to be good [44].

*Pain-related partner behavior (Multidimensional Pain Inventory: MPI-D; [45]; German version [46]).* We used part 2 of the MPI-D. Since it had originally been designed to measure pain-related partner behavior in chronic pain samples, we adapted the instruction for the current study to assess the partner's behavior in acute pain situations such as head-, tooth-, joint- or muscle ache. Part 2 of the MPI-D consists of three scales: (a) *Punishing responses* (3 items, e.g., *"Expresses anger at me"*), (b) *Distracting responses* (3 items, e.g., *"Encourages me to work on a hobby"*), and (c) *Solicitous responses* (5 items, e.g., *"Asks me how he can help"*). The items are rated on a 7-point Likert scale ranging from 0 (*never*) to 6 (*very often*). The MPI-D subscales have good consistency (Cronbach's α = .75 - .93), reliability ($r_{tt}$ = .70 - .89) and validity [46].

## 3. Experimental design

The experiment consisted of 15 trials of tonic heat pain stimulation with simultaneous picture presentation. The inter-trial-interval varied randomly between 15 and 35 s.

**3.1 Heat pain stimulation.** All heat stimuli were applied with a 30 x 30 mm Peltier element-based advanced thermal stimulator (ATS) thermode (Pathway Model Cheps, Medoc Ltd, 2005, Ramat Yishai, Israel) placed on the thenar of the non-dominant hand.

In each experimental trial, tonic heat stimulation started from a baseline temperature of 32˚C, with temperature increasing at a rate of 1˚C/s until the individually adjusted target temperature was reached and maintained for 54 s. Then, temperature returned to baseline at a rate of 8˚C/s.

The intensity of the experimental tonic heat pain stimuli was individually adjusted such that (a) the intensity of the tonic heat pain stimulus was rated as about 10 on a visual analogue scale (VAS, 0–20 units), and that (b) pain intensity would not be rated below 8 on the VAS during the 50 s adjustment trial stimulation. To determine the individual stimulus intensity, a two-step approach was followed: (a) The heat pain threshold (HPT) was determined using the method of limits. Following three practice trials, five HPT trials were run. HPT was defined as the mean of the last three trials; (b) The temperature for the tonic heat pain stimulation was determined by a series of tonic heat trials with an increase in stimulation temperature. Each trial during the adjustment phase lasted 50 s, started at a baseline temperature of 32˚C which increased at a rate of 1˚C/s until reaching the target temperature. In the first trial, the target temperature was set at 1˚C below HPT. In each subsequent trial, the temperature of the tonic heat stimulus was increased by 0.5˚C. During the 50 s tonic stimulation, the participants rated the pain intensity continuously using the VAS. Subsequently, the temperature returned to baseline at a rate of 8˚C/s. After an inter-trial-interval of 30 s, participants started the next heat stimulus by pressing the Enter button. The calibration procedure was terminated when pain intensity was rated at about 10 and not less than 8 on the VAS units.

**3.2 Picture stimuli.** Five categories of pictures were presented to investigate social-affective pain modulation: (a) partner neutral faces (from now on referred to as 'partner'), (b) standard stranger neutral faces ('neutral'), (c) standard stranger happy faces ('happy'), (d) standard stranger angry faces ('angry'), and (e) standard neutral non-social objects ('objects'). To systematically vary affective valence, we used happy, angry and neutral faces of strangers. Angry faces can be considered as particularly ecologically valid negative social stimulus as they can convey a punishing response of a person.

We decided against manipulating the affective valence of the partners' faces for reasons of undue burden on participants (increase in number of trials) and their partners (need to provide additional photos expressing happiness and anger) and difficulties in ensuring the validity of the target emotional expressions. Nevertheless, the picture categories allowed to compare the effects of neutral partner, neutral stranger, and object pictures, and, in addition, the differential effects of strangers' neutral, happy, and angry faces. Hence, conclusions can be drawn both regarding the role of the social content as well as affective valence.

Happy, neutral and angry faces of strangers were taken from the Karolinska Directed Emotional Faces picture set [47]. We chose 15 male frontal pictures for each facial expression (neutral, happy, angry) based on the best hit rates of emotion recognition and the highest intensity scores ([48]; KDEF image codes: neutral: AM11NES, AM31NES, AM13NES, AM10NES, AM01NES, AM07NES, AM06NES, AM25NES, AM08NES, AM14NES, AM18NES, AM35NES, AM04NES, BM12NES, BM16NES; happy: AM07HAS, AM25HAS, AM32HAS, AM17HAS, AM22HAS, AM23HAS, AM05HAS, AM12HAS, AM01HAS, AM04HAS, AM08HAS, AM09HAS, AM11HAS, AM16HAS, AM31HAS; angry: AM10ANS, AM11ANS, AM17ANS, AM09ANS, AM30ANS, AM29ANS, AM24ANS, AM19ANS, AM26ANS, AM28ANS, AM31ANS, BM12ANS, BM18ANS, AM13ANS, AM06ANS). The 562 x 762 pixel pictures were colored and in portrait format with a display size of 14.4 x 19.6 cm. Examples are shown in Fig 1A.

In addition, there was a set of 30 neutral pictures of household objects taken from the International Affective Picture System (IAPS [34]; image codes neutral objects: 7000, 7002, 7004, 7006, 7009, 7010, 7020, 7025, 7030, 7034, 7035, 7040, 7041, 7050, 7052, 7056, 7059, 7080, 7090, 7175, 7150, 7179, 7185, 7186, 7217, 7233, 7235, 7705, 7950). These 1024 x 768 pixel sized pictures were colored and in landscape format (display size: 25.9 x 19.6 cm).

Fifteen pictures of the partner were taken prior to the experiment either at home or in the laboratory under standardized conditions. Partners were instructed to wear a black, grey or white T-shirt, and to look straight into the camera without smiling. If the pictures were taken at home, the men were requested to take the photos in the absence of their partners. All pictures were checked for deviations from a neutral expression. When partners' pictures were not considered to be neutral, new pictures were taken. Partner pictures were matched in size to the standard faces (562 x 762 pixels).

In each trial, six pictures were randomly selected from one picture category. There were three trials per picture category in a pseudo-random order such that the same picture category was not shown more than twice in consecutive trials (see Fig 1A). Furthermore, each picture could not be presented more than twice within and across trials.

**3.3 Trial structure.** Experimental design and trial structure are illustrated in Fig 1A and 1B. Each experimental trial consisted of a tonic heat stimulus with a duration of 54 s. During each tonic heat stimulus six pictures of one of the five picture categories were presented, each picture for 5 s. Presentation of the pictures 1–3 is further referred to as picture viewing time 1 (PVT1; total duration: 15 s), the presentation of the pictures 4–6 as PVT2 (total duration: 15 s).

**3.4 Outcome measures.** *3.4.1 Pain intensity, picture valence and arousal ratings. Pain intensity* was rated three times per trial on a visual analogue scale (VAS) presented on the PC

**Fig 1. a. Experimental design.** Each picture category was presented three times (block 1–3). In each block, the order of the five picture categories was randomized for every participant. Inter-trial-intervals (ITI) varied between 15 and 35 s. Image codes of the schematically displayed pictures (from left to right): AM10ANS, AM31NES, AM07HAS, instead of the picture of the partner the picture AM02NES is presented, due to picture rights, 7009. **b. Trial structure.** The figure shows the time course of one experimental trial. (A) Psychophysiological data were recorded continuously and later segmented (BL: baseline, 5s before temperature rising, pic1-pic6: 6 5 s intervals during picture presentation, PVT: picture viewing time), (B) heat pain stimulation: starting from a baseline temperature of 32°C, temperature increased at a rate of 1°C/s until reaching the individually adjusted stimulation temperature. The target temperature was held constant for 54 s and then returned to baseline temperature at a rate of 8°C/s, (C) pain intensity ratings on a visual analogue scale (VAS): VAS were shown on the screen for 8 s and participants were asked to rate pain intensity prior to picture viewing (VAS t1), and after each picture viewing time (VAS t2: after pic1-pic3; VAS t3: after pic4-pic6); (D) picture presentation and ratings: valence and arousal ratings were obtained using the self-assessment manikin (SAM). ITI: 15-35s.

monitor ranging from no pain (0) to worst pain imaginable (20) within a time window of 8 s: (1) VAS t1—immediately when the target experimental temperature was reached and prior to picture viewing, (2) VAS t2 –after the end of PVT1, and (3) VAS t3 –after PVT2.

*Valence and arousal*. At the end of each trial, when the heat pain stimulus had returned to baseline temperature, participants were asked to rate the valence and the arousal of the six pictures presented during the trial. Valence and arousal ratings were obtained using computerized versions of the well-established Likert self-assessment manikins (SAM; [49]; valence: 1 = very unpleasant; 9 = very pleasant; arousal: 1 = very calm; 9 = very arousing). Participants made their ratings by moving the cursor from a mid-scale starting point by pressing the 'left'/'right' arrow keys.

*3.4.2 Psychophysiological data*. All psychophysiological signals were recorded continuously with a QuickAmp 72 Amplifier using BrainVision Recorder software (Brain Products GmbH, 2008) and sampled at 1000 Hz. A ground electrode was placed on the right clavicle. Except when asked to rate, participants were instructed not to move their hands. Markers were recorded such that the data segments for BL, PVT1 and PVT2 could be extracted offline. All other recordings over time were not further considered for analysis.

*Skin conductance level* (SCL*)*. For SCL recording, a galvanic skin response adaptor (GSR sensor, Brain Products GmbH) and two Ag/AgCl Electrodes (diameter: 10 mm) filled with TD-246 isotonic paste (EASYCAP GmbH, Herrsching, Germany) were used. The electrodes were placed on the thenar eminence of the dominant hand after having cleansed the hand with water. The signal was filtered online with a 100 Hz low pass filter. The mean SCL (mS) for 500 ms epochs was calculated. Data of two participants had to be excluded from further analysis due to technical problems during recording.

*Corrugator electromyography (*Corrugator EMG*)*. Activity of the M. corrugator supercilii was measured using a bipolar reording with two Ag/AgCl electrodes (diameter: 5mm) filled with electrolyte-gel (SuperVisc, Easycap GmbH). After cleansing the skin with alcohol, the electrodes were placed above the left eyebrow according to EMG recording guidelines [50]. The corrugator EMG was filtered online with a 10Hz high pass filter. The EMG signal was filtered offline with a 50 Hz notch filter and visually inspected for artefacts. The EMG signals were rectified and integrated using contour-following integrators with a time constant of 80 ms [36]. Mean EMG activity (μV) was calculated for 500 ms epochs. Artefact-containing 500ms epochs were excluded from further analysis.

*Heart Rate (HR)*. Heart rated was measured using an electrocardiogram (ECG) with two pre-gelled AG/AgCl electrodes (Megro, Wesel, Germany). One electrode was placed on the right clavicle and the other one on the lowest left rib. ECG signals were filtered online with a 100 Hz low pass filter and a 0.5Hz high pass filter to avoid baseline shifts in the ECG [51]. The signal was visually inspected for artefacts and R-Wave detection errors. For HR analysis, beat-to-beat intervals were calculated and transformed into beats per minute for every data point and averaged across 500 ms epochs. All artefact-containing 500 ms epochs were excluded from further analysis. The data of one participant had to be excluded from further analysis due to technical problems during recording.

## 4. Procedure

When a female participant was considered as eligible, her partner was sent an email with a standardized instruction on how to take the necessary neutral portrait photos. Partners who were not able to take the photos at home were offered to be photographed in the laboratory. All partners provided informed consent for the use of their photos in the study. Female participants were invited via email to fill out the relevant questionnaires online.

The experiment took place in a psychophysiology lab at the local university. The experimental chamber was equipped with a video camera, a microphone and speakers to allow communication with and monitoring of the participants from an adjacent control room. The experimental run was controlled by *Presentation* software (Neurobehavioral Systems, Inc., Version 14.2) from the adjacent control room. Participants received written instructions throughout the experiment.

Participants were seated comfortably in a chair with armrests in front of a PC desk with a monitor (distance: 0.5 m), a mouse and a computer keyboard. Next, participants were informed about the experiment and gave their informed consent. Then, the thermode was attached to the non-dominant hand with a velcro strap and the electrodes for the physiological recordings were attached.

Heat pain thresholds were determined and participants were familiarized with the SAM ratings. Next, the experimental temperature was adjusted individually. Immediately prior to the experimental trials, the participants rated their current happiness in the relationship. The actual experiment lasted approximately 20 min. After the experiment, the electrodes and the thermode were removed, participants were debriefed and had the opportunity to ask questions.

## 5. Data preprocessing and statistical analysis

**5.1 Data preprocessing.** *Ratings*. For each picture category and each rating time (VAS t1, VAS t2, VAS t3), the mean pain intensity was calculated as the average over the three trials. For the correlational analyses, we defined partner pain modulation (partner-PM) as the difference in pain intensity between trials with neutral and partner faces (neutral–partner) at VAS t3, thus allowing to account for individual differences in pain experiences. Accordingly, partner-PM scores greater than 0 reflect a pain-relieving effect of the partner faces when compared to neutral faces. Mean valence and arousal for each picture category were calculated by averaging the ratings over the three trials per picture category.

*Psychophysiological data*. Preprocessing and aggregation of the psychophysiological data was done with customized Matlab programs. To reduce inter-individual differences, all physiological raw data were individually z-transformed [$z_i = (x_i - M_i)/SD_i$]. As we used tonic heat pain stimulation, we were interested in psychophysiological responses during the heat pain stimulation. Therefore, after standardization (z-scores), we computed mean values for seven 5 s intervals per trial (BL, pic1-pic6) which were further collapsed over the three trials per picture category. Finally, averages for PVT1 (i.e., mean of pic1 to pic3) and for PVT2 (i.e., mean of pic4 to pic6) were computed, yielding mean values for PVT1, PVT2, and BL (see Fig 1B).

**5.2 Statistical analysis.** All statistical analyses were calculated using IBM SPSS Statistics (version IBM 20.0.0, Armonk, NY) and *p*-values were set to .05.

*Analyses of variance*. We used the GLM procedure to conduct the ANOVAs to analyze differences in valence and arousal, pain intensity modulation and psychophysiological correlates depending on picture category and time. If the assumption of sphericity was violated, Greenhouse-Geisser corrected degrees of freedom (*df*) were used, yet, the nominal *df* are reported. For *F*-tests, partial eta-squared ($\eta^2$) were calculated as effect sizes. Significant interactions were followed up by post-hoc Bonferroni adjusted estimated marginal mean tests between picture categories for VAS t1-t3 using the COMPARE function of the GLM procedure [52]. Cohen's *d* for repeated measures ($d_{rm}$; [53]) was calculated as effect size for mean comparisons.

For differences in valence and arousal between pictures, two one-way repeated measures ANOVAs were conducted with the factor PICTURE (5: partner, happy, neutral, angry, objects).

The pain modulation by picture category was analyzed using a two-way 3 x 5 repeated measures ANOVA with the factors TIME (VAS t1, 2, 3) and PICTURE (partner, happy, neutral, angry, objects).

For the psychophysiological correlates, two-way 3 x 5 repeated measure ANOVAs with the factors TIME (BL, PVT1, PVT2) and PICTURE (partner, happy, neutral, angry, objects) were conducted for SCL, corrugator EMG and HR, respectively. In addition, due to different time courses of HR changes during PVT1 and PVT2, difference scores between the first and last 5 s segments of PVT1 and PVT2 were computed, thus HR difference scores above 0 indicate HR deceleration between the early and late phase of picture viewing.

*Correlational analyses*. The relationship between the partner-PM and perceived valence and arousal of the partner face, perceived partner support, relationship satisfaction and pain-related catastrophizing was determined by computing Pearson's product-moment correlations or point-biserial correlations. In order to determine the unique relationship of a variable with partner-PM, partial correlations were calculated, whenever a variable correlated significantly both with partner-PM and another of the considered variables.

## Results

### 1. Valence and arousal ratings for each picture category

Results regarding valence and arousal ratings are summarized in Table 2.

The one-way repeated measure ANOVA for valence ratings revealed a significant main effect for PICTURE ($F(4,112) = 71.21$, $p < .001$, $\eta^2 = .72$). Partner and happy faces were perceived as equally pleasant ($p = 1.00$, $d = 0.34$) and both evoked more pleasure than neutral faces and objects ($ps < .001$, $ds > 1.32$) which elicited similar pleasure ($p = .063$, $d = 0.55$). Angry faces were perceived to be most unpleasant ($ps < .001$, $ds > 1.79$).

The one-way repeated measure ANOVA for arousal ratings also found a main effect of PICTURE ($F(4,112) = 21.7$, $p < .001$, $\eta^2 = .44$). Angry faces and partner faces were most arousing and did not differ significantly ($p > 0.999$, $d = 0.04$). Angry faces elicited significantly more subjective arousal than happy faces ($p = .008$, $d = 0.59$), neutral faces and objects ($ps < .001$, $ds > 1.23$). Partner faces were more arousing than objects and neutral faces ($ps < .001$, $ds > 1.11$) which did not differ significantly in subjective arousal ($p = .211$, $d = 0.40$). The difference in arousal between partner and happy faces had a medium effect size but did not reach statistical significance ($p = .058$, $d = 0.53$). Happy faces also elicited significantly more arousal than neutral faces ($p = .018$, $d = 0.78$) and objects ($p < .001$, $d = 1.00$).

### 2. Modulation of pain intensity by pictures

*Experimental temperature and pain intensity*. The mean temperature used in the experiment was $M = 43.72°C$ ($SD = 1.53$; range 40.5–46.5°C). The mean pain intensity rating at t1 (i.e., before any picture viewing) in the very first experimental trial was $M = 11.1$ ($SD = 3.98$).

*Pain modulation during picture viewing*. The 5 x 3 repeated measure ANOVA of pain intensity ratings during picture presentation revealed a main effect of PICTURE ($F(4,112) = 7.93$,

**Table 2. Valence and arousal ratings ($M$, $SD$) for the pictures.**

| Picture | partner | | happy | | angry | | neutral | | objects | |
|---|---|---|---|---|---|---|---|---|---|---|
| | *Mean* | *SD* | *Mean* | *SD* | *Mean* | *SD* | *Mean* | *SD* | *Mean* | *SD* |
| Valence | 7.43[a] | 1.35 | 6.99[a] | 1.19 | 2.87[b] | 1.16 | 4.93[c] | 1.14 | 5.48[c] | 1.07 |
| Arousal | 4.28[ab] | 2.37 | 3.15[a] | 1.65 | 4.37[b] | 2.25 | 2.07[c] | 0.98 | 1.67[c] | 1.04 |

[abc] Significant differences between categories are indicated by different superscript.

$p < .001$, $\eta^2 = 0.22$), TIME ($F(2,56) = 7.94$, $p = .005$, $\eta^2 = 0.22$), and a significant PICTURE x TIME interaction effect ($F(8,224) = 12.57$, $p < .001$, $\eta^2 = 0.31$). Results are displayed in Fig 2.

At baseline, there were no significant differences in pain intensity ratings (VAS t1) between the five different picture categories ($Ms = 7.68–8.24$, $ps > .37$, $ds < 0.17$).

At t2, pain intensity was lower when viewing partner rather than angry faces, neutral faces, and objects [$Ms$ ($p$-value) = 5.98 vs. 7.44 ($< .001$), 7.46 (.005), 7.60 (.001), $ds > 0.49$]. Viewing happy faces ($M = 6.72$) also led to lower pain intensity compared to objects ($M = 7.60$; $p = .012$, $d = 0.34$). The difference between partner ($M = 5.97$) and happy faces ($M = 6.72$) was not significant ($p = .102$, $d = 0.28$).

At t3, the pattern of differences in pain intensity depending in picture category was similar to t2. Pain intensity was significantly lower when viewing partner rather than angry faces, neutral faces and objects [$Ms$ ($p$-value) = 6.16 vs. 8.17 ($< .001$), 8.03 (.001), 8.14 ($< .001$), $ds > 0.60$]. Happy faces as compared to angry faces and objects lowered perceived pain intensity [$Ms$ ($p$-value) = 6.86 vs. 8.17 (.013), 8.14 (.011), $ds > 0.41$]. Again, pain intensity while viewing partner ($M = 6.16$) and happy faces ($M = 6.86$) did not differ significantly ($p = .162$, $d = 0.24$).

## 3. Psychophysiological responses during tonic heat pain and picture viewing

The time course of SCL, corrugator EMG and HR is displayed in Fig 3. To analyze the effects of picture viewing on pain-related psychophysiological responses, 3 x 5 repeated measure ANOVAs with the factors TIME (BL, PVT1, PVT2) and PICTURE (partner, happy, neutral, angry, objects) were conducted.

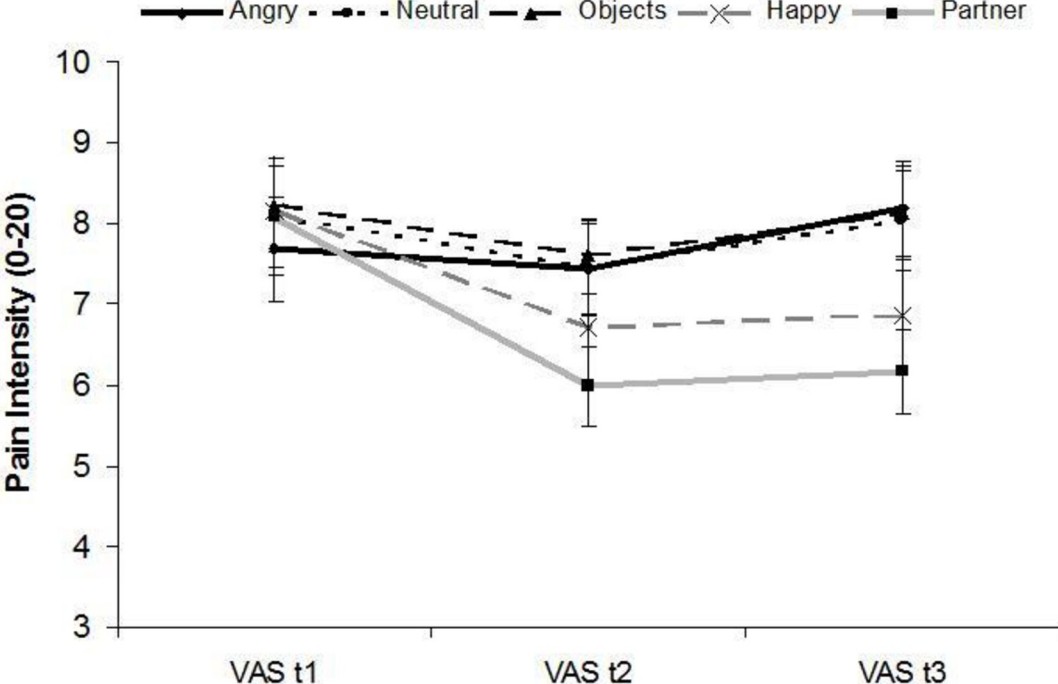

**Fig 2. Pain intensity ratings during tonic heat pain stimulation.** Ratings on a visual analogue scale (VAS; M, SE) before picture viewing (VAS t1), after picture viewing time (PVT) 1 (VAS t2) and after PVT 2 (VAS t3).

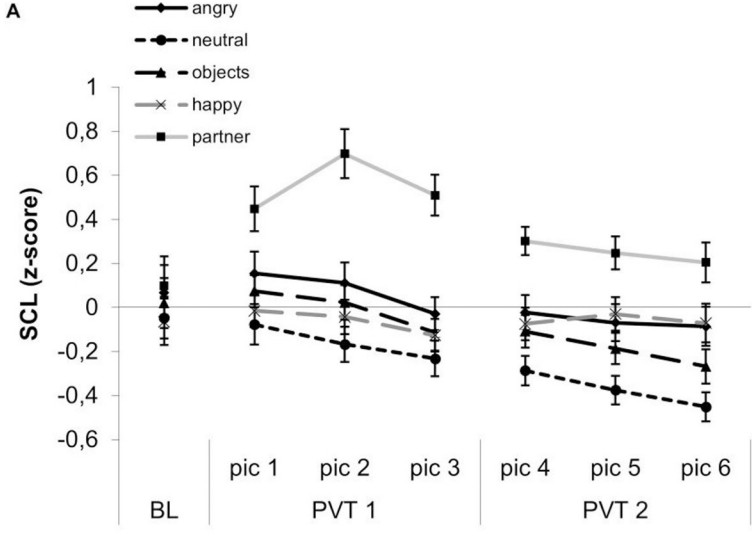

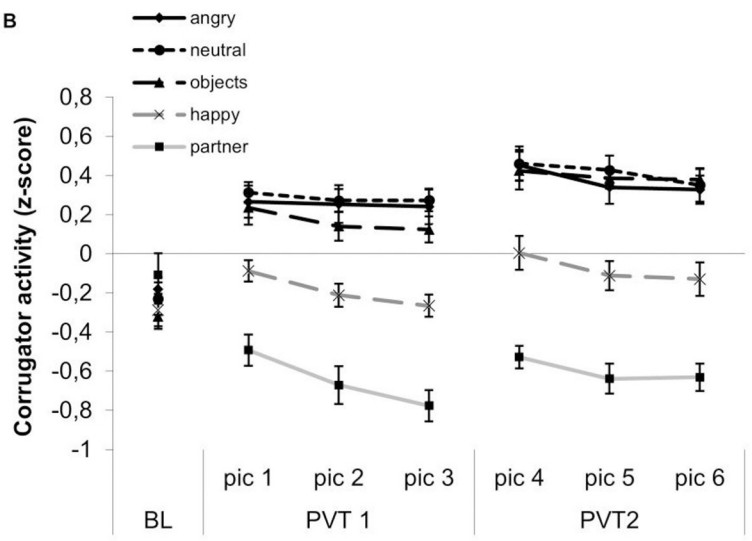

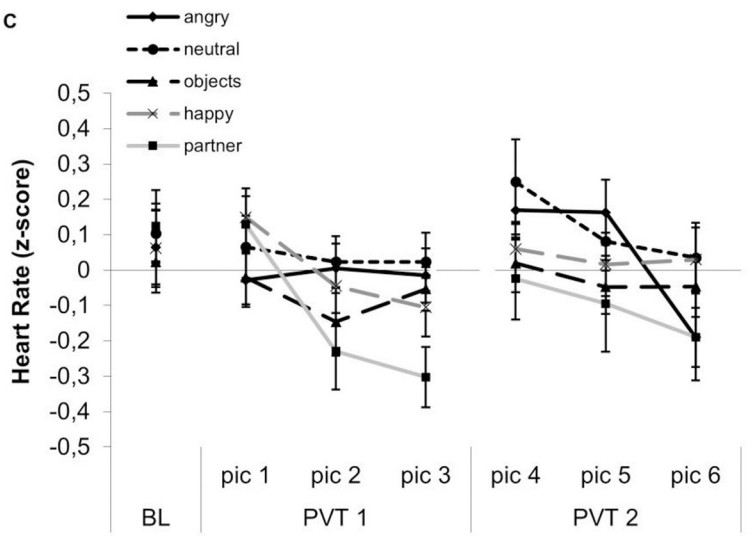

**Fig 3. Time course of psychophysiological responses during tonic heat pain combined with picture-viewing.** 5s averages of (A) skin conductance level (SCL), (B) corrugator activity, and (C) heart rate (HR); BL = baseline, PVT 1 = picture viewing time 1 with pictures 1–3 (pic 1–3), PVT2 = picture viewing time 3 with pictures 4–6 (pic 4–5).

*SCL*. The analysis of SCL during picture viewing while being exposed to tonic heat pain revealed a main effect of PICTURE ($F_{(4,104)}$ = 6.03, $p < .001$, $\eta^2$ = .19), and a significant PIC-TURE x TIME interaction ($F_{(8,208)}$ = 4.63, $p = .002$, $\eta^2$ = .15) (see Fig 4A). There were no significant BL differences in SCL depending on picture category (-.07 < *M*s .10, *p's* = 1, *d*s < 0.26; see Fig 3A). During PVT 1, viewing partner faces elicited the highest SCL compared to all other picture categories (PVT1: *d*s = 1.01–1.59). Similarly, during PVT2, partner faces were associated with higher SCL compared to objects (*d* = 1.16) and neutral faces (*d* = 1.77) (see Fig 4A for contrasts). Further, during PVT2, SCL was larger for happy compared to neutral faces (*d* = 0.89). Finally, during PVT2, SCL was higher when viewing partner compared to happy faces (*d* = 0.82), and when viewing angry compared to neutral faces (*d* = 0.84), though not reaching statistical significance (*p* = .091 partner vs. happy; *p* = .056 angry vs. neutral).

*Corrugator EMG*. The 3 x 5 ANOVA for the corrugator EMG revealed significant main effects of PICTURE ($F_{(4,112)}$ = 24.55, $p < .001$, $\eta^2$ = .45) and TIME ($F_{(2,56)}$ = 19.17, $p < .001$, $\eta^2$ = .41), and a significant PICTURE x TIME interaction ($F_{(8,224)}$ = 20.86, $p < .001$, $\eta^2$ = .43). As illustrated in Fig 4B, at BL, there were no differences in corrugator activity between the picture categories (*M*s = -.11 –-.32, *p*s > .31, *d*s < .43). During PVT1 and PVT2, when viewing partner faces corrugator activity was significantly lowest in contrast to all other picture categories (PVT1: *d*s = 1.17–2.73, PVT2: *d*s = 1.36–3.00). Furthermore, during PVT1 and PVT2, corrugator activity was significantly lower for happy faces when compared to all other picture categories except partner faces (PVT1: *d*s = 0.97–1.51, PVT2: *d*s = 1.13–1.26).

*HR*. The 3 x 5 ANOVA showed no main effects for TIME or PICTURE and no interaction effect (*p*s > .149, $\eta^2$ < .07). As illustrated in Fig 4C, this indicates that HR did not differ between picture categories, at least when HR was averaged for the total PVTs. However, as illustrated in Fig 3C, HR changed differentially over the course of PVT1 and PVT2 depending on picture category. To delineate this time course, we computed difference scores between the first and last 5 s segment of PVT1 and PVT2, respectively. Difference scores greater than 0 reflect a deceleration in HR from the beginning to the end of picture viewing.

We conducted a 2 x 5 repeated measures ANOVA with the factors PVT (PVT1 vs 2) and PICTURE CATEGORY (5) and the HR change score as dependent variable. The ANOVA yielded a significant interaction between PICTURE CATEGORY and PVT ($F_{(4,108)}$ = 3.56, $p = .009$, $\eta^2$ = .12), but no significant main effects for PVT (*p* = .774) and PICTURE CATE-GORY (*p* = .122). During PVT1, HR deceleration was greater when viewing partner than angry and neutral faces and objects [*M*s (*p*-values) = .43 vs. -.01 (.004), .04 (.009), and .03 (.002); *d*s = 0.67–0.89] and more pronounced when viewing happy (*M* = .25) compared to angry faces (*M* = -.01, *p* = .049, *d* = 0.56). For PVT1, no other comparison reached significance (*p*s >.107, *d*s < 0.47). During PVT2, HR deceleration was greater when viewing angry faces (*M* = .36) compared to happy faces (*M* = .03, *p* = .012, *d* = .63). For PVT2, no other comparison reached significance (*p*s > .050, *d*s < 0.56).

## 4. Correlations between partner-PM, partner-related and psychological variables

Correlation coefficients for the association between the experimental measures of partner-PM, and partner-related variables are summarized in Table 1. Neither perceived partner support (DCI) nor relationship satisfaction (RSA) were associated with partner-PM. Catastrophizing

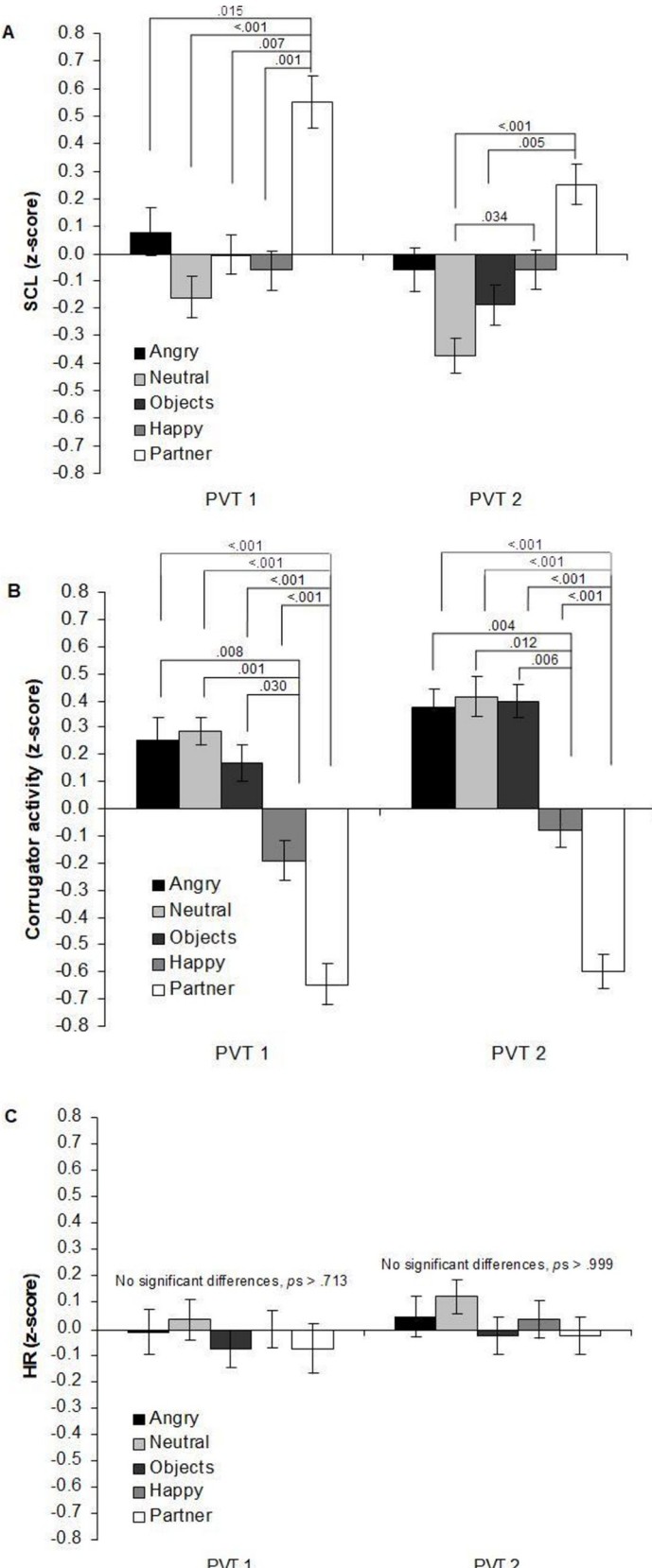

**Fig 4. Average z-scores of psychophysiological responses for the five picture categories during picture viewing.** Descriptive statistics (*M*, *SE*) for (A) skin conductance level (SCL), (B) corrugator activity, and (C) heart rate (HR) during picture viewing time (PVT) 1 and 2. *p*-values refer to the difference between picture categories.

correlated significantly with partner-PM, i.e., the higher the disposition for pain-related catastrophizing, and particularly helplessness, the stronger the pain-relieving effect of viewing partner rather than neutral pictures. Moreover, subjective arousal of the partner pictures significantly correlated with pain-related helplessness (PCS) and solicitous pain-related partner behavior (MPI-D). Interestingly, the association between pain-related helplessness and partner-PM was no longer significant when controlled for arousal ($r_{\text{PCS-PM.Arousal}} = .32$, $p = .095$). Yet, when controlled for pain-related helplessness, the association between arousal and partner-PM remained significant ($r_{\text{PM-Arousal.PCS}} = .41$, $p = .032$).

## Discussion

The main aim of the present study was to examine the pain modulating effect of pictures varying in affective quality and social content. Pain modulation was evaluated both at the subjective and the physiological level. An additional aim was to identify variables such as induced affect, catastrophizing, perceived partner support and relationship satisfaction that might account for social pain modulation.

As intended, valence and arousal ratings of the pictures demonstrate successful induction of positive and negative affect using pictures varying along social and affective dimensions. Partner faces were rated as highly pleasant and moderately arousing, angry faces as highly unpleasant and moderately arousing. Happy faces were rated to be as pleasant as partner faces, but were less arousing. Valence of neutral faces and objects was rated as in between happy and angry faces and low arousing.

Pain modulating effects emerged both at the level of subjective experience and at the level of psychophysiological responses. Pain intensity was reduced most when viewing partner and, somewhat less pronounced, when viewing happy faces. Viewing angry and neutral faces and neutral objects had no effect on reported pain intensity. Partner faces elicited highest SCL and lowest corrugator activity. Happy faces were associated with lower corrugator activity than neutral faces, angry faces and objects, but not as low as the partner faces. SCL in response to happy faces was higher than for neutral ones. The influence of picture viewing on pain-related HR response revealed a more complex pattern (see below).

The obtained pain modulation by partner pictures at the level of subjective experience is consistent with previous reports of pain modulating effects when viewing partners rather than strangers and objects [15, 16], or when viewing acquaintances compared to completing a distraction task [17]. Our findings extend previous reports in that, unlike in previous studies [15–17], we carefully controlled the partners' facial expression to be neutral, did not rely on personal pictures provided by the participants [17] and did not instruct participants to think about the partner when the photograph was presented [17]. Importantly, pain modulation was not limited to subjective report. The diminished corrugator activity closely matched the pain relief at the level of subjective report. In light of eyebrow squeezing (i.e., an increase in corrugator activity) constituting a core characteristic of the facial expression of pain [54], the lower corrugator activity suggests a diminished behavioral response to pain when viewing partner pictures. In accordance with the MPT [18], this pain-alleviating effect might be accounted for by the partner pictures inducing a positive affective state as suggested by them being rated as highly pleasant. It is also consistent with previous accounts attributing the pain-relieving effect of partner faces to the activation of feelings of being loved and supported [16].

As the partners in our study had a neutral facial expression, our findings demonstrate that the pain-relieving effect of partners does not require facially expressed happiness. Indeed, as suggested by the small effect size, neutral partner faces diminished subjective pain intensity somewhat more than happy faces, although this difference did not reach statistical significance. At the level of pain-related facial activity, however, partner faces were associated with the lowest corrugator activity. The induction of intense positive feeling by neutral partner faces may be partly accounted for by the familiarity effect [55]. Evidence suggests that familiar and unfamiliar faces are processed differentially. More specifically, familiar faces are more consistently recognized and socially evaluated than unfamiliar ones. Mileva and colleagues [56] interpret this familiarity effect as a shift from image to person rating, i.e., becoming familiar with a face implies gathering knowledge about the individual facial variability and having to rely less on changing facial expressions. This familiarity effect can account for partners' faces being perceived as highly positive stimuli despite the neutral facial expression. To our knowledge, however, this moderating effect of familiarity has not been systematically tested so far, including, for example, also partners' faces expressing negative emotions such as anger. Furthermore, we believe that our findings also suggest that a picture of the partner with a neutral facial expression is sufficient to activate associated mental representations [57]. Unlike in previous studies [15, 16], the partners had not accompanied the participants when coming to the lab and were not specifically instructed to think about the partner [17], thus underlining the strength of the priming effect. Indeed, partners' pictures were associated with a greater increase in SCL which likely reflects the strength of the activation of approach motivation [58].

Regarding to strangers' faces, happy, but not angry faces had a pain modulating effect as shown by pain self-report and corrugator activity. Few studies have investigated pain modulation using positive (e.g., happy) as well as negative facial expressions (e.g., angry, sad, fearful). Sad [58–60] or angry [21] faces have been associated with increased pain intensity, whereas viewing happy faces reduced pain intensity [58–60], except in one study reporting enhanced pain [21]. From the perspective of the Emotional Control of Nociception model (ECON, [59]), the rather consistent finding of pain attenuating effect of happy faces, as in the present study, is only partially accounted for. Our finding is consistent with the assumption of the ECON model that pictures' valence determines the direction of pain modulation. It is, however, more difficult to reconcile with the assumption that the modulating influence of valence depends on the level of arousal. Emotional faces compared to IAPS pictures are typically perceived as less arousing [22, 60]. In the present study, partner, happy and angry faces were all rated as moderately arousing. The theory of social signal value of emotion expression [61] could explanain for the finding that happy (and partner) faces had a pain modulating effect despite being moderately arousing. One of its basic assumptions is that an expressed emotion provides information not only about the expressor, but also about the situation, and vice versa, and that context information can influence the perceived emotion. From this perspective, happy (and partner faces) might be a safety cue and of particular importance in a threatening pain context [1, 23, 62, 63]. This approach is consistent with the finding that brain regions related to safety-signals are only activated during pain relief when partner pictures are presented [15]. As postulated by Krahé and colleagues [1], when experiencing pain, one's social interactions may serve as predictive signals of contextual threat or safety, thus modulating the salience of pain. In the PVT paradigm used here, partners may function as particularly powerful safety signals as also suggested by the obtained SCL increase.

Unlike for partner and happy faces, angry faces had no pain modulatory effect. Angry faces were rated as highly aversive and moderately arousing. Indeed, corrugator EMG and SCL for angry faces were not significantly different from neutral faces and objects. Especially the lack of a SCL increase suggests that avoidance motivation was not strongly activated [58]. When

compared to unpleasant pictures having a pain-enhancing effect [e.g., 37], the angry faces were perceived as much less arousing. In fact, the arousal ratings of the angry faces were comparable to pictures of loss (M = 4.89) which were also found not to significantly enhance pain in a previous study [59]. Hence, the lack of a pain-modulating effect of angry faces is consistent with predictions of the ECON model regarding arousal [59]. As noted earlier, there is only one study yielding a pain enhancing effect of angry faces [21]. Yet, in this study [21] both positive (happy) and negative (fear, anger) faces were associated with increased pain, thus making the findings difficult to interpret and compare. Indeed, from an evolutionary psychology approach [64, 65], one might argue that pain suppression rather than expression is beneficial as a stranger's anger may indicate a threat of being exploited [12, 13, 65].

A major aim of the present study was to investigate pain modulation by affective and social cues not only using subjective report, but also psychophysiological correlates. As a measure both of facial pain expression [54] and unpleasantness of stimuli [30], corrugator activity was reduced when viewing partner and happy pictures, with partner faces eliciting the strongest decrease. This is likely to reflect the partners' faces being perceived as highly pleasant as well as the reduced experienced pain intensity. Despite this similarity in corrugator activity between happy and partner faces, there were also differences. Only partner faces elicited higher physiological arousal as indicated by SCL. This corresponds well with the arousal ratings of happy and partner faces in our study, as the latter were rated as more arousing. It is also consistent with prior research [66–69] in which partner pictures and neutral unfamiliar control pictures were compared to same sex friends, parents, or siblings, thereby controlling for familiarity. In these studies, familiar as compared to unfamiliar faces generally led to higher autonomic responses as measured for example by SCL. Interestingly, among familiar faces, it is not unspecific arousal that differentiates between partners and other familiar persons (i.e., parent, sibling, friend). Specifically, the higher subjective arousal co-occurs with a specific pattern of physiological changes (HR, skin conductance, activity of the zygomaticus muscle), thus supporting romantic love being an unique intense positive emotion which overlaps, but can be distinguished from other positive emotions [67, 69]. Correspondingly, romantic love is associated with a certain brain activation pattern which is distinct from other positive emotional states [66]. This unique pattern of intense positive emotion and the concomitant activation of approach motivation elicited by partner pictures is reflected by differences in SCL and corrugator EMG between partner and happy pictures, and can account for the stronger effect of partners rather than happy strangers on pain modulation.

Results regarding HR were more complex. We did not observe a modulating influence of the pictures on pain-related changes in HR when focusing on the average HR during PVT1 and PVT2. Yet, when the difference in HR between the first and last 5 s during PVTs was considered, partner faces were associated with greater HR deceleration. Sustained HR deceleration reflecting pain-relief has been observed in previous studies [19, 37], thus masking a possible HR acceleration which has been demonstrated in response to the presentation of faces of loved ones without concomitant pain stimulation [32]. Due to the presentation of three pictures in a row during each PVT, it was not possible to disentangle overlapping time courses of HR changes in response to the emotional cues as has been demonstrated in pure picture viewing paradigms [58]. Clearly, our findings can only be interpreted as preliminary and require replication focusing on the course of HR changes in response to each picture with a sufficiently long inter-stimulus-interval.

We were also interested in elucidating relationship and psychological factors that might contribute to partner-PM. Neither relationship duration, relationship satisfaction nor perceived partner support were significantly related with partner-PM. This could be due to the restricted variance in quality and duration of partnership and/or restrictions in the chosen

questionnaires measuring support and satisfaction. While relationship satisfaction was relatively high in our sample, it was comparable to the level reported for different non-clinical samples of married and dating couples [40, 70, 71]. Similarly, stress-related partner support was comparable to the norms as provided for individuals of the same age [46]. Possibly, partner's support behavior as assessed by questionnaire may be a better predictor when there is an actual opportunity for partners to interact with participants. Further, the experience of being supported may become more important when pain has become chronic [72].

Interestingly, pain-related catastrophizing, particularly helplessness, and arousal were associated with partner-PM. In our sample, the average level of pain catastrophizing was not in the clinical range (i.e., below the cut-off of 30 [38]), but comparable to a clinical sample of low back pain patients [39]. Participants who reported higher levels of helplessness and rated their partners' pictures as more arousing experienced greater pain relief when viewing their partners' pictures. Partial correlation analyses suggest a mediating effect of arousal for the relation between pain-related helplessness and partner-PM. This finding is intriguing as it corresponds well with assumptions of the communal model of pain catastrophizing [73]. According to this model, pain catastrophizing is a coping response by which a person's expression of pain elicits support and empathic responses from other. This could account for the greater pain relief when viewing partners' picture in participants with high levels pain-related catastrophizing. The mediating effect of arousal is compatible with the notion that arousal indexes the degree of activation of a motivational system [58], i.e., the pain diminishing effect of a partner when a person engages in pain catastrophizing is stronger when the activation of approach motivation is high.

The main limitation of our study is the reduced generalizability as our sample consisted of a comparatively small sample of young, healthy women in a close long-term relationship. Several reasons led us to focus on a female sample with the aim of controlling for various potential confounding factors. Gender effects have been documented when using facial expression as stimuli, for observers of facial expressions, and for experimenters [74]. Moreover, male facial expressions of anger modulate the startle response more strongly than female facial expressions, possibly interacting with the gender of the participant [75]. Also, there is evidence that women are better at judging emotions from facial expressions [76]. In experimental settings, male participants report less pain and tolerate more pain in the presence of female experimenters, thus pointing to complex interactions between the gender of the participant, the gender of social cues and the experimenter [77, 78]. To avoid these confounding effects, and to ensure better comparability with previous studies also relying on female participants [15, 16], the sample included only female participants. The sample was representative regarding relationship quality. Specifically, relationship satisfaction was comparable to different non-clinical samples of married and dating couples [40, 70, 71]. Similarly, self-reported stress-related partner support was well within the normal range of support levels to be expected in this age group [44]. Even though the sample size might be considered somewhat limited, it is comparable or even larger than in previous experimental studies investigating the influence of social cues on pain [15–17]. The within-subject and randomized design is a strength of the study as well as the z-score standardization to adjust for interindividual differences in psychophysiological responding.

Another limitation is that we did not use a full factorial design including different emotional expressions of partners. Aside from concerns about imposing greater burden on the participants by having to considerably increase the number of trials and the duration of the experiment, partners were expected not to be used to reliably and repeatedly showing distinct facial expressions of anger and happiness, thus seriously threatening the validity of such pictures. One might also argue that pictures have limited ecological validity, for example in

comparison to video sequences or the partner being present in the laboratory. Yet, the picture viewing paradigm we used ensures standardization and systematic variation of affect and social content.

Despite these limitations, our findings extend what is known about social and affective factors modulating pain. We show that partners' pain-relieving effect does not depend on a positive facial expression of emotion such as happiness. Indeed, neutral partner pictures alleviated pain even more than pictures of happy faces of strangers at the level of pain-related facial activity. Clinically, our results imply that just looking at pictures of one's partner when undergoing acute painful procedures may have a hypoalgesic effect which can be expected not to be limited to subjective report, but to be paralleled by changes in autonomic arousal and pain-related facial activity. Interestingly, in one study, when tested in a group setting, chronic pain patients' acute pain report was reduced when viewing pictures of loved ones which they themselves had chosen among their personal photos, thus possibly inducing a particularly positive emotional response [79]. Being able to rely on a neutral picture of a significant other may be especially useful in situations when the patient is in pain or undergoing a painful procedure and significant others such as the partner or parents are overly concerned [80] or little supportive [81]. In such cases, viewing partner or parent pictures might in fact be more helpful than the actual interaction.

## Acknowledgments

Authors wish to thank Annina Klapper for her help with data collection, Laurens Berthold for his help with data analysis, Matthias Sperl for helpful comments on an earlier version of this manuscript, and all participants and their partners for their support.

## Author Contributions

**Conceptualization:** Katrin Hillmer, Christiane Hermann.

**Data curation:** Katrin Hillmer.

**Formal analysis:** Katrin Hillmer.

**Investigation:** Katrin Hillmer.

**Methodology:** Katrin Hillmer.

**Project administration:** Katrin Hillmer.

**Supervision:** Christiane Hermann.

**Visualization:** Katrin Hillmer.

**Writing – original draft:** Katrin Hillmer, Judith Kappesser.

**Writing – review & editing:** Katrin Hillmer, Judith Kappesser, Christiane Hermann.

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
