## [Decision Letter · Decision Letter 0]

19 Jan 2021

PONE-D-20-37976

Pain modulation by your partner: an experimental investigation from a social-affective perspective

PLOS ONE

Dear Dr. Kappesser,

Thank you for submitting your manuscript to PLOS ONE. I have now received comments from to experts. As you will note, both reviewers were rather positity about your study but also raised a number of concerns that should be fully addressed before the manuscript is ready for publication. You should pay particular attention to the issues regarding the interpretation of data and the need of some clarification in the methods and results section.

We look forward to receiving your revised manuscript.

Kind regards,

José A Hinojosa, Ph.D.

Academic Editor

PLOS ONE

Journal Requirements:

Reviewers' comments:

Reviewer's Responses to Questions

**Comments to the Author**

1. Is the manuscript technically sound, and do the data support the conclusions?

Reviewer #1: Partly

Reviewer #2: Partly

2. Has the statistical analysis been performed appropriately and rigorously? 

Reviewer #1: Yes

Reviewer #2: Yes

3. Have the authors made all data underlying the findings in their manuscript fully available?

Reviewer #1: No

Reviewer #2: Yes

4. Is the manuscript presented in an intelligible fashion and written in standard English?

Reviewer #1: No

Reviewer #2: Yes

5. Review Comments to the Author

Reviewer #1: This manuscript describes an interesting experiment aimed to examine, by psychophysiological and subjective pain intensity measures, the influence of the emotional meaning conveyed by different facial expressions (partner and stranger people) on pain outcomes while participants are undergoing a paradigm involving the presence of tonic heat pain. The potential links between these measures and other affective and partner-related variables (support and relationship satisfaction) were also explored. Although the research questions are of interest to the field and the study has potential to make a relevant contribution, from my point of view, some conclusions are not supported by the data or at least it should be more extendedly discussed. Moreover, there are other concerns about the manuscript that would need to be addressed. I will detail below these points in a way that I hope is useful for the authors.

1) My main concern is about the interpretation of the data related to the role of a familiarity effect for explaining differences between partners and strangers on pain perception (see Abstract and Discussion). Note that both partner and happy facial expression of strangers reduced equally pain perception reported by the participants. Thus, part of their conclusion as well as interpretations related to this are not supported by the data, and maybe even misleading. Authors indicate “Therefore, familiar faces such as partner faces may - independent of the actual facial expression displayed - bring to mind a wide range of intimate interactions. This familiarity effect corresponds well with the accounts explaining the pain-relieving effect of partner faces by the activation of being loved and supported.” However, happy faces displayed by strangers led to a similar reduction in pain perception. On the other hand, the fact that angry faces did not produce an increase in pain perception should be also more specifically addressed. A careful and more precise interpretation of the present data is strongly recommended.

2) Regarding the functional meaning reflected by psychophysiological measures I would have liked to see a bit more introduction and discussion on the mechanisms underlying the effect of facial expressions; in other words, how are SCL, corrugator activity and heart rate are involved in the modulation of pain perception by emotional facial expressions (partners and strangers).

3) Methods: Experimental design and stimuli.

Experimental design section deserves a more detailed explanation. Although the Figure 1 includes the most important aspect associated with the experimental task, the explanation of it on the main text is hard to follow. As a single example, I recommend moving the last paragraph included in the Picture Stimuli section (lines 213-217) and the Procedure section to the Experimental design section in order to make it more understandable.

Apart from that, there are many relevant aspects related to the facial expression stimuli that are missing:

-How do participants rate the level pain perceived from the thermal stimulator? They had to write with her dominant hand on the VAS? If so, how movement artifacts were removed from the SCL recording?

-Physical features of facial expression stimuli and objects are missing. Did they cropped in an oval to remove non-facial features? Did they present in black and white color? It would be recommended to include an example of each type of facial expression in the Figure 1. A thoroughly description of it should be added.

-Please, include the image codes from facial stimuli taken from Karolinska Directed Faces picture set.

-Information about the procedure is very confused. It is unclear the description on the presentation order of each block of facial expressions stimuli across different participants.

-Valence and arousal ratings made by the participants at the end of each trial is incomplete too. It is unclear whether participants were presented again with faces in order to be rated or not.

-Captions of figures are missing

I short, Experimental design and Stimuli sections (pictures and heat pain stimulation) should be edited and thoroughly revised.

4) Another important concern is the inclusion of only women in the experimental sample and the important limitations associated of it. As the authors recognize in the line 547, “emotion differs depending on whether male or female emotional faces are used…” This single comment is not enough and a more elaborated explanation about this important limitation should be added. In my opinion, the current data might lead to different kinds of misunderstanding. I am very sorry that I cannot be more positive at this point. I think that the research question is interesting, the paradigm is appropriate. If the authors could collect data from a larger a male sample all these aspects will be not confounded. I would be very enthusiastic about this study.

5) Several passages in the Discussion section should be much more detailed and elaborated. Whereas SCL effects for partners’ faces are clearly exposed, functional mechanisms underlying such results are not provided. This idea should be more developed, and an overview of the previous literature is required in order to interpret the functional meaning and implications of these results. The explanation included in lines 517-518 has a speculative feel with respect to data obtained in the present study.

6) The manuscript is generally written in an understandable and clear manner, yet it would benefit from review by a native English speaker.

7) Line 446: Table 1 should be changed by Table 3.

Reviewer #2: PONE-D-20-37976

This study aims to explore social-affective influence over acute pain perception using different psychophysiological measures (corrugator EMG, SCL and HR). The main limitation of this study is the reduced generalizability, because the sample is small (less than 30 subjects) and consists of young healthy women (mean=23.34, SD=2.89) in a close long-term relationship (69.9% had been in the partnership longer than two years). However, the results are interesting, but I have some questions regarding the methods that might help to improve the manuscript:

INTRODUCTION

Page 4, line 70. Add in parentheses the acronym that is later used (line 73).

METHODS

Page 11, line 248. “The pain intensity ratings for VAS t1-t3 were averaged”. Did the authors check before averaging that there were no statistically significant differences?

Page 11, lines 249-251. “Partner pain modulation (partner-PM) was defined as the difference in pain intensity between neutral and partner faces (neutral-partner) for VAS t3”. Why the authors did not use the average?

Pages 12-13, lines 259-287. The authors should clarify which analysis measure they used for each psychophysiological variable. In the case of SCL and corrugator EMG, did the authors measure mean amplitude?

Page 14, line 309. It should be added that all participants also evaluated the valence and arousal of the stimuli.

Page 14, lines 314-317. Again, it is not very clear to me how the authors have analyzed the physiological data. They have performed averages per trial (groups of 6 images of a single category), but the presentation of each image must generate a physiological response (which, in the case of SCL, is very slow).

Page 14, line 318. Add the SPSS version used.

RESULTS

Page 16, lines 370-371. If during the individual adjustment of the intensity of the painful stimulus, the VAS had to be above 10 points (50%), why is the range so wide (2.13-16.20) in the experimental condition?

Page 17, lines 376-377. It is evident that no significant differences will be found in the pain intensity ratings at t1, because it is before the presentation of the images (or the BL of each trial). I suggest it be written as on line 399.

Page 18, line 401. “The effect was less pronounced at PVT2”. Did the authors find statistically significant differences?

Page 18, lines 414-421. HR deceleration analysis should be described in the methods section. Why did the authors not do this type of analysis with the other two physiological measures?

Page 20, lines 453-456. The authors should describe in the methods section how controlled for arousal and helplessness in correlational analyses.

DISCUSSION

Page 23, line 541. If the authors have performed partial correlations, they should describe in the methods section.

6. PLOS authors have the option to publish the peer review history of their article (what does this mean?). If published, this will include your full peer review and any attached files.

Reviewer #1: No

Reviewer #2: No

---

## [Author Response · Author response to Decision Letter 0]

10 May 2021

Please find our point-by-point response to the reviewers’ comments below. 

General Comments

1. Data availability: We would like to point out that the data underlying the statistical data analysis as well as the research materials can in general be made available. With regard to the pictures, we have now included the picture codes for those pictures which were taken from published picture sets (KDEF, IAPS). With regard to the partner pictures, the participants and their partner only gave consent to use the pictures as part of the experiment. In order to comply with data protection laws, we cannot make these freely available, especially since there is no possibility to anonymize the pictures. The data which were collected in this experiment are not freely and directly available because the original approval by the local ethics committee of the Faculty of Psychology and Sports Science, Justus-Liebig University Giessen, Germany (#2010-0018) and the informed consent form signed by the participants did not include such direct access. However, the anonymized data will be made available to interested researchers upon request. We now include a footnote in the manuscript indicating how to gain access to the data. 

2. English writing style: We have carefully checked the manuscript for proper English writing style and, accordingly, have changed the wording throughout the paper. 

Comments Reviewer #1: 

1) My main concern is about the interpretation of the data related to the role of a familiarity effect for explaining differences between partners and strangers on pain perception (see Abstract and Discussion). Note that both partner and happy facial expression of strangers reduced equally pain perception reported by the participants. Thus, part of their conclusion as well as interpretations related to this are not supported by the data, and maybe even misleading. Authors indicate “Therefore, familiar faces such as partner faces may - independent of the actual facial expression displayed - bring to mind a wide range of intimate interactions. This familiarity effect corresponds well with the accounts explaining the pain-relieving effect of partner faces by the activation of being loved and supported.” However, happy faces displayed by strangers led to a similar reduction in pain perception. 

We thank the reviewer for critically commenting on our conclusions regarding the role of the familiarity effect for pain modulation. We agree that our conclusion that familiarity is responsible for the pain modulating effect of neutral partner faces has been somewhat misleading. It is important to note that the partner faces were associated with the lowest corrugator activity and increased SCL. Moreover, even at the subjective level, the effect size suggests that partner faces are somewhat more effective in reducing pain. It is likely that the familiarity effect contributes to perceiving the partner pictures as intensely positive stimuli despite their neutral facial expressions and that - considering the similarities to happy faces - this intense positive emotion is crucial for pain modulation. We now discuss the results in more detail. Accordingly, we amended the sentence in the Abstract and the paragraph in the Discussion. The sentence in the Abstract now reads:

Abstract:

Results

Partner and happy faces reduced self-reported pain intensity and corrugator activity, the latter being lowest when viewing partner faces as compared to all other picture categories. As corrugator activity is indexing stimulus unpleasantness and a core feature of the facial pain expression, this physiological pattern matches well with the subjective ratings. …

Discussion

… Within the context of socially adaptive behavior, they particularly underline the social signal value of emotion and attachment figures. 

The amended paragraphs in the Discussion now reads:

As the partners in our study had a neutral facial expression, our findings demonstrate that the pain-relieving effect of partners does not require facially expressed happiness. Indeed, as suggested by the small effect size, neutral partner faces diminished subjective pain intensity somewhat more than happy faces, although this difference did not reach statistical significance. At the level of pain-related facial activity, however, partner faces were associated with the lowest corrugator activity. The induction of intense positive feeling by neutral partner faces may be partly accounted for by the familiarity effect (55). Evidence suggests that familiar and unfamiliar faces are processed differentially. More specifically, familiar faces are more consistently recognized and socially evaluated than unfamiliar ones. Mileva and colleagues (56) interpret this familiarity effect as a shift from image to person rating, i.e. becoming familiar with a face implies gathering knowledge about the individual facial variability and having to rely less on changing facial expressions. This familiarity effect can account for partners’ faces being perceived as highly positive stimuli despite the neutral facial expression. To our knowledge, however, this moderating effect of familiarity has not been systematically tested so far, including, for example, also partners’ faces expressing negative emotions such as anger. Furthermore, we believe that our findings also suggest that a picture of the partner with a neutral facial expression is sufficient to activate associated mental representations (57). Unlike in previous studies (15, 16), the partners had not accompanied the participants when coming to the lab and were not specifically instructed to think about the partner (17), thus underlining the strength of the priming effect. Indeed, partners’ pictures were associated with a greater increase in SCL which likely reflects the strength of the activation of approach motivation (58).

On the other hand, the fact that angry faces did not produce an increase in pain perception should be also more specifically addressed.

We agree with the reviewer that we should have addressed this finding in more detail. We now describe the assumptions of the Emotional Control of Nociception (ECON) model and compare the results for the angry faces with previous data based on the ECON model. Moreover, we take into account the physiological response into account. We would like to point out that there has been only one other study evaluating the pain modulatory effect of angry faces (21), which, however, reported a pain-enhancing effect of positive and negative emotional facial expressions. We expanded the part about angry faces. It now reads: 

With regard to strangers’ faces, happy, but not angry faces had a pain modulating effect as shown by pain self-report and corrugator activity. Few studies have investigated pain modulation using positive (e.g. happy) as well as negative facial expressions (e.g., angry, sad, fearful). Sad (58–60) or angry (21) faces have been associated with increased pain intensity, whereas viewing happy faces reduced pain intensity (58-60), except in one study reporting enhanced pain (21). From the perspective of the Emotional Control of Nociception model [ECON, (59)], the rather consistent finding of pain attenuating effect of happy faces, as in the present study, is only partially accounted for. Our finding is consistent with the assumption of the ECON model that pictures’ valence determines the direction of pain modulation. It is, however, more difficult to reconcile with the assumption that the modulating influence of valence depends on the level of arousal. Emotional faces compared to IAPS pictures are typically perceived as less arousing (22, 60). In the present study, partner, happy and angry faces were all rated as moderately arousing. The theory of social signal value of emotion expression (61) could offer an explanation for the finding that happy (and partner) faces had a pain modulating effect despite being moderately arousing. One of its basic assumptions is that an expressed emotion provides information not only about the expressor, but also about the situation, and vice versa, and that context information can influence the perceived emotion. From this perspective, happy (and partner faces) might be a safety cue and of particular importance in a threatening pain context (1, 23, 62, 63). This corresponds well with the finding that brain regions related to safety-signals are only activated during pain relief when partner pictures are presented (15). As postulated by Krahé and colleagues (1), when experiencing pain, one’s social interactions may serve as predictive signals of contextual threat or safety, thus modulating the salience of pain. In the PVT paradigm used here, partners may function as particularly powerful safety signals as also suggested by the obtained SCL increase. 

Unlike for partner and happy faces, angry faces had no pain modulatory effect. Angry faces were rated as highly aversive and moderately arousing. Indeed, corrugator EMG and SCL for angry faces were not significantly different from neutral faces and objects. Especially the lack of a SCL increase suggests that avoidance motivation was not strongly activated (58). When compared to unpleasant pictures having a pain-enhancing effect (e.g. 37), the angry faces were perceived as much less arousing. In fact, the arousal ratings of the angry faces were comparable to pictures of loss (M = 4.89) which were also found not to significantly enhance pain in a previous study (59). Hence, the lack of a pain-modulating effect of angry faces is consistent with predictions of the ECON model regarding arousal (59). As noted earlier, there is only one study yielding a pain enhancing effect of angry faces (21). Yet, in this study (21) both positive (happy) and negative (fear, anger) faces were associated with increased pain, thus making the findings difficult to interpret and compare. Indeed, from an evolutionary psychology approach (64, 65), one might argue that pain suppression rather than expression is beneficial as a stranger’s anger may indicate a threat of being exploited (12, 13, 65). 

A careful and more precise interpretation of the present data is strongly recommended.

We agree with the reviewer and have changed the Discussion thoroughly according to your and the second reviewer’s comments and suggestions. Since the discussion is completely revised and numerous additions have been made, we refrain from copying the whole Discussion section here in this revision letter. Rather, we would like to refer the reviewer to the revised manuscript. 

2) Regarding the functional meaning reflected by psychophysiological measures I would have liked to see a bit more introduction and discussion on the mechanisms underlying the effect of facial expressions; in other words, how are SCL, corrugator activity and heart rate are involved in the modulation of pain perception by emotional facial expressions (partners and strangers).

We agree with the reviewer that we did not sufficiently address the functional meaning of the psychophysiological measures that were used to elucidate pain modulation. The autonomic (SCL, HR) measures were included based on studies evaluating the ECON model of pain modulation as well as based on the Motivational Priming Theory (MPT) and the literature on aversive conditioning. Activity of the corrugator was measured because, according to MPT, an increase correlates strongly with the perceived unpleasantness of a stimulus. Moreover, enhanced corrugator activity underlies eyebrow squeezing which is a core element of the facial expression of pain. Clearly, the psychophysiological response pattern observed for affective and social modulation of pain represents the combined response to the pain and the emotional stimuli. Nonetheless, the observed psychophysiological response allows conclusion as to the degree of activation of the sympathetic nervous system (SCL), of the autonomic nervous system in general (HR), and the activation of avoidance motivation (‘unpleaseantness’) and pain expression (EMG corrugator). We now have expanded the relevant paragraphs in the abstract, the introduction and the discussion. 

Abstract:

Results

Partner and happy faces reduced self-reported pain intensity and corrugator activity, the latter being lowest when viewing partner faces as compared to all other picture categories. As corrugator activity is indexing stimulus unpleasantness and a core feature of the facial pain expression, this physiological pattern matches well with the subjective ratings. Neutral objects, neutral and angry faces had no effect on pain self-report, although angry faces were rated as highly negative. Partner faces also led to increased skin conductance being an index of motivational activation, and heart rate deceleration which may reflect increased sensory intake. Partner-related pain modulation was primarily related to perceived arousal of the partner’s picture, i.e. the intensity of the activation of approach motivation, and pain-related catastrophizing.

Introduction: 

It would be particularly meaningful to demonstrate that social cues modulate pain not only at the level of subjective report, but also at the level of pain-related psychophysiological responses. Due to being a highly relevant biological stimulus, pain induces autonomic arousal, as indicated by higher skin conductance level (SCL) and heart rate (HR) (26, 27), i.e. a physiological response pattern which is characteristic for fight-flight behavior (21, 28). Moreover, pain stimuli are associated with increased activity of the corrugator muscle, which is consistent both with findings of eyebrow squeezing being a core element of the facial expression of pain (29), and corrugator activity correlating with perceived aversiveness of stimuli (30). Viewing pleasant and unpleasant scenes as well as emotional facial expressions is associated with higher SCL as compared to neutral pictures (18, 30, 31). Moreover, pleasant picture elicit less corrugator activity (30, 32), whereas unpleasant increase corrugator activity (20, 33–36). HR responses to emotional pictures vary depending on valence, yet, are complex due to differences in time course (31–33). Little is known about HR responses when participants are exposed to both pain and emotional stimuli. Interestingly, there is evidence that pain-related HR acceleration is attenuated while viewing pleasant pictures (37). 

Discussion: One particular focus when changing the Discussion was the interpretation of psychophysiological measures. The changed paragraphs are referred to in the response to your comment #5 (see below).

3) Methods: Experimental design and stimuli. Experimental design section deserves a more detailed explanation. Although the Figure 1 includes the most important aspect associated with the experimental task, the explanation of it on the main text is hard to follow. As a single example, I recommend moving the last paragraph included in the Picture Stimuli section (lines 213-217) and the Procedure section to the Experimental design section in order to make it more understandable.

In short, Experimental design and Stimuli section (pictures and heat pain stimulation) should be edited and thoroughly revised.

We apologize that the Method section was somewhat difficult to follow. Following your advice, the method section was thoroughly revised. The experimental design section is completely restructured. In section ‘3. Experimental design’, we first only summarize the design in one sentence, but then explain in more detail how the experimental pain stimuli (3.1) were determined and which pictures (3.2) were used as stimuli. In section 3.3., we included a section describing the trial structure in which we explain the experimental task in more detail so that it is easier to follow. Aside from the original Figure 1 (now 1b), Section 3.3 now also entails a new Figure (1a) which illustrates the experimental design including examples of the picture categories, information about the pictures and picture category randomization across trials and blocks. Following Section 3.3., the Outcome measures (3.4), Procedure (4.) and Data preprocessing and statistical analysis (5.) are reported. Specifically, the following changes were made: 

3. Experimental design

The experiment consisted of 15 trials of tonic heat pain stimulation with simultaneous picture presentation. The inter-trial-interval varied randomly between 15 and 35 s. 

3.1 Heat Pain Stimulation

All heat stimuli were applied with a 30 x 30 mm Peltier element-based advanced thermal stimulator (ATS) thermode (Pathway Model Cheps, Medoc Ltd, 2005, Ramat Yishai, Israel) placed on the thenar of the non-dominant hand. 

In each experimental trial, tonic heat stimulation started from a baseline temperature of 32°C, with temperature increasing at a rate of 1°C/s until the individually adjusted target temperature was reached and maintained for 54 s. Then, temperature returned to baseline at a rate of 8°C/s.

The intensity of the experimental tonic heat pain stimuli was individually adjusted such that (a) the intensity of the tonic heat pain stimulus was rated as about 10 on a visual analogue scale (VAS, 0-20 units), and that (b) pain intensity would not be rated below 8 on the VAS during the 50 s adjustment trial stimulation. To determine the individual stimulus intensity, a two-step approach was followed: (a) The heat pain threshold (HPT) was determined using the method of limits. Following three practice trials, five HPT trials were run. HPT was defined as the mean of the last three trials; (b) The temperature for the tonic heat pain stimulation was determined by a series of tonic heat trials with an increase in stimulation temperature. Each trial during the adjustment phase lasted 50 s, started at a baseline temperature of 32°C which increased at a rate of 1°C/s until reaching the target temperature. In the first trial, the target temperature was set at 1°C below HPT. In each subsequent trial, the temperature of the tonic heat stimulus was increased by 0.5°C. During the 50 s tonic stimulation, the participants rated the pain intensity continuously using the VAS. Subsequently, the temperature returned to baseline at a rate of 8°C/s. After an inter-trial-interval of 30 s, participants started the next heat stimulus by pressing the Enter button. The calibration procedure was terminated when pain intensity was rated at about 10 and not less than 8 on the VAS units.

3.2 Picture Stimuli

Five categories of pictures were presented to investigate social-affective pain modulation: (a) partner neutral faces (from now on referred to as ‘partner’), (b) standard stranger neutral faces (‘neutral’), (c) standard stranger happy faces (‘happy’), (d) standard stranger angry faces (‘angry’), and (e) standard neutral non-social objects (‘objects’). To systematically vary affective valence, we used happy, angry and neutral faces of strangers. Angry faces can be considered as particularly ecologically valid negative social stimulus as they can convey a punishing response of a person. 

We decided against manipulating the affective valence of the partners’ faces for reasons of undue burden on participants (increase in number of trials) and their partners (need to provide additional photos expressing happiness and anger) and difficulties in ensuring the validity of the target emotional expressions. Nevertheless, the picture categories allowed to compare the effects of neutral partner, neutral stranger, and object pictures, and, in addition, the differential effects of strangers’ neutral, happy, and angry faces. Hence, conclusions can be drawn both with regard to the role of the social content as well as affective valence.

Happy, neutral and angry faces of strangers were taken from the Karolinska Directed Emotional Faces picture set (47). We chose 15 male frontal pictures for each facial expression (neutral, happy, angry) based on the best hit rates of emotion recognition and the highest intensity scores (48) . The 562 x 762 pixel pictures were colored and in portrait format with a display size of 14.4 x 19.6 cm. Examples are shown in Figure 1a.

In addition, there was a set of 30 neutral pictures of household objects taken from the International Affective Picture System [IAPS (34)] . These 1024 x 768 pixel sized pictures were colored and in landscape format (display size: 25.9 x 19.6 cm). 

Fifteen pictures of the partner were taken prior to the experiment either at home or in the laboratory under standardized conditions. Partners were instructed to wear a black, grey or white T-shirt, and to look straight into the camera without smiling. If the pictures were taken at home, the men were requested to take the photos in the absence of their partners. All pictures were checked for deviations from a neutral expression. When partners’ pictures were not considered to be neutral, new pictures were taken. Partner pictures were matched in size to the standard faces (562 x 762 pixels).

In each trial, six pictures were randomly selected from one picture category. There were three trials per picture category in a pseudo-random order such that the same picture category was not shown more than twice in consecutive trials (see Figure 1a). Furthermore, each picture could not be presented more than twice within and across trials.

3.3 Trial structure

Experimental design and trial structure are illustrated in Figures 1a and b. Each experimental trial consisted of a tonic heat stimulus with a duration of 54 s. During each tonic heat stimulus six pictures of one of the five picture categories were presented, each picture for 5 s. Presentation of the pictures 1-3 is further referred to as picture viewing time 1 (PVT1; total duration: 15 s), the presentation of the pictures 4-6 as PVT2 (total duration: 15 s).

------

Insert Figure 1a,b about here

------

4. Apart from that, there are many relevant aspects related to the facial expression stimuli that are missing: 

a) How do participants rate the level pain perceived from the thermal stimulator? They had to write with her dominant hand on the VAS? If so, how movement artifacts were removed from the SCL recording? 

Again, we apologize that we had omitted such relevant details. Participants had to press the “left”/ “right” cursor keys to move the cursor on the screen when rating pain intensity, valence and arousal. In-between ratings, they were instructed not to move their hands. Though recorded, the psychophysiological data during the rating time window was discarded offline from further analysis. It is now more clearly described that markers were recorded which allowed to extract the relevant data segments. We also would like to note that the SCL electrodes were placed on the thenar eminence of the hand, thus not interfering with finger movements. We added this information in section 3.4 “Outcome measures”. The relevant sentences now read:

3.3 Outcome measures

3.3.1 Pain intensity, picture valence and arousal ratings

Participants made their ratings by moving the cursor from a mid-scale starting point by pressing the ‘left’/’right’ arrow keys.

3.3.2 Psychophysiological data

All psychophysiological signals were recorded continuously with a QuickAmp 72 Amplifier using BrainVision Recorder software (Brain Products GmbH, 2008) and sampled at 1000 Hz. A ground electrode was placed on the right clavicle. Except when asked to rate, participants were instructed not to move their hands. 

b) Physical features of facial expression stimuli and objects are missing. Did they cropped in an oval to remove non-facial features? Did they present in black and white color? It would be recommended to include an example of each type of facial expression in the Figure 1. A thoroughly description of it should be added. Please, include the image codes from facial stimuli taken from Karolinska Directed Faces picture set.

Again, we apologize that we have not provided sufficient information about the pictures which were used. As mentioned in our response to your comment #3, we included Figure 1 b in which one example of each picture category is displayed. 

Figure 1a: Experimental design. Each picture category was presented three times (block 1-3). In each block, the order of the five picture categories was randomized for every participant. Inter-trial-intervals (ITI) varied between 15 and 35 s. Image codes of the schematically displayed pictures (from left to right): AM10ANS, AM31NES, AM07HAS, instead of the picture of the partner the picture AM02NES is presented, due to picture rights, 7009.

In addition, we now provide the image codes not only for the Karolinska Directed Faces picture set but also for the IAPS images. Providing the codes made us realize that we made a mistake when describing the number of pictures in each category. Unlike stated in the original manuscript, there were 15 (and not 18) pictures per face category and 30 pictures in the neutral objects’ picture pool. 

The amended paragraph describing the picture stimuli now reads:

Happy, neutral and angry faces of strangers were taken from the Karolinska Directed Emotional Faces picture set (47). We chose 15 male frontal pictures for each facial expression (neutral, happy, angry) based on the best hit rates of emotion recognition and the highest intensity scores (48) . The 562 x 762 pixel pictures were colored and in portrait format with a display size of 14.4 x 19.6 cm. Examples are shown in Figure 1a.

In addition, there was a set of 30 neutral pictures of household objects taken from the International Affective Picture System [IAPS (34)] . These 1024 x 768 pixel sized pictures were colored and in landscape format (display size: 25.9 x 19.6 cm).

Fifteen pictures of the partner were taken prior to the experiment either at home or in the laboratory under standardized conditions. Partners were instructed to wear a black, grey or white T-shirt, and to look straight into the camera without smiling. If the pictures were taken at home, the men were requested to take the photos in the absence of their partners. All pictures were checked for deviations from a neutral expression. When partners’ pictures were not considered to be neutral, new pictures were taken. Partner pictures were matched in size to the standard faces (562 x 762 pixels).

c) Information about the procedure is very confused. It is unclear the description on the presentation order of each block of facial expressions stimuli across different participants.

We thank the reviewer for bringing this to our attention. As stated in our response to comments #4a,b, we hope that Figure 1a,b now illustrate more clearly the presentation order of each block of facial expression stimuli across participants.

d) Valence and arousal ratings made by the participants at the end of each trial is incomplete too. It is unclear whether participants were presented again with faces in order to be rated or not.

Again, thanks for bringing this up. As stated earlier, we have revised the whole Experimental design section. Specifically, we now clarify that valence and arousal ratings were made based on a second presentation of the pictures originally presented in this trial. The relevant sentences in 3.3 now reads:

Valence and arousal. At the end of each trial, when the heat pain stimulus had returned to baseline temperature, participants were asked to rate the valence and the arousal of the six pictures presented during the trial. Valence and arousal ratings were obtained using computerized versions of the well-established Likert self-assessment manikins (SAM; 49; valence: 1 = very unpleasant; 9 = very pleasant; arousal: 1 = very calm; 9 = very arousing). Participants made their ratings by moving the cursor from a mid-scale starting point by pressing the ‘left’/’right’ arrow keys.

e) Captions of figures are missing 

To the best of our knowledge, the captions of all figures (Figure 1-4) had been included in the manuscript. We have revised the figure captions such that they provide more details. The figure captions are placed beneath each figure.

4) Another important concern is the inclusion of only women in the experimental sample and the important limitations associated of it. As the authors recognize in the line 547, “emotion differs depending on whether male or female emotional faces are used…” This single comment is not enough and a more elaborated explanation about this important limitation should be added. In my opinion, the current data might lead to different kinds of misunderstanding. I am very sorry that I cannot be more positive at this point. I think that the research question is interesting, the paradigm is appropriate. If the authors could collect data from a larger a male sample all these aspects will be not confounded. I would be very enthusiastic about this study. 

We fully agree that our reasoning for including only women needs to be elaborated in more detail. Gender was restricted solely in an attempt to maximally control for confounding factors: Gender effects have been documented (though not consistently) in studies in which the gender of the facial expressions used as stimuli was varied. In general, male faces expressing anger modulate the startle response more strongly, with a possible interaction with the gender of the participant (Paulus et al., 2014). Gender effects have also been demonstrated or observers of facial expressions. For example, women are better at judging emotions from facial expressions (Keogh, 2014; McClure, 2000). Lastly, interactions between the gender of participants and experimenters are well documented. For example, men report less experienced pain and have a higher pain tolerance in the presence of a female rather than a male experimenter (Aslaksen et al., 2007; Kállai et al., 2004). In light of these findings, gender of the participants would possibly have impacted many stages of our experimental design (e.g., male participants – female participants; female face pictures vs. male face pictures) and, eventually the results of our experiment and. To avoid this confounding effect of gender, we decided to include only women as participants. Also, we aimed at extending previous findings with regard to partner-related pain modulation by carefully testing the differential role of the affect and social content of the pictures. Hence, it seemed important to us to maximize comparability with these previous studies relying on female samples. 

Following your advice, we now explain sample selection in more detail in the limitation section in the Discussion. It now reads: 

Several reasons led us to focus on a female sample with the aim of controlling for various potential confounding factors. Gender effects have been documented when using facial expression as stimuli, for observers of facial expressions, and for experimenters (74). Moreover, male facial expressions of anger modulate the startle response more strongly than female facial expressions, possibly interacting with the gender of the participant (75). Also, there is evidence that women are better at judging emotions from facial expressions (76). In experimental settings, male participants report less pain and tolerate more pain in the presence of female experimenters, thus pointing to complex interactions between the gender of the participant, the gender of social cues and the experimenter (77, 78). To avoid these confounding effects, and to ensure better comparability with previous studies also relying on female participants (15, 16), the sample included only female participants.

5) Several passages in the Discussion section should be much more detailed and elaborated. Whereas SCL effects for partners’ faces are clearly exposed, functional mechanisms underlying such results are not provided. This idea should be more developed, and an overview of the previous literature is required in order to interpret the functional meaning and implications of these results. The explanation included in lines 517-518 has a speculative feel with respect to data obtained in the present study. 

We are grateful for bringing this to our attention. As stated earlier, we have completely revised the Discussion section and some parts have been newly added. The discussion was restructured and we now address the results for the psychophysiological measure in much more detail, including their functional meaning. We now also carefully relate and interpret our results based on theoretical models and empirical findings based on these. As an example of all the changes we have made, the section discussing the psychophysiological findings now reads as follows:

A major aim of the present study was to investigate pain modulation by affective and social cues not only using subjective report, but also psychophysiological correlates. As a measure both of facial pain expression (54) and unpleasantness of stimuli (30), corrugator activity was reduced when viewing partner and happy pictures, with partner faces eliciting the strongest decrease. This is likely to reflect the partners’ faces being perceived as highly pleasant as well as the reduced experienced pain intensity. Despite this similarity in corrugator activity between happy and partner faces, there were also differences. Only partner faces elicited higher physiological arousal as indicated by SCL. This corresponds well with the arousal ratings of happy and partner faces in our study, as the latter were rated as more arousing. It is also consistent with prior research (66–69) in which partner pictures and neutral unfamiliar control pictures were compared to same sex friends, parents, or siblings, thereby controlling for familiarity. In these studies, familiar as compared to unfamiliar faces generally led to higher autonomic responses as measured for example by SCL. Interestingly, among familiar faces, it is not unspecific arousal that differentiates between partners and other familiar persons (i.e. parent, sibling, friend). Specifically, the higher subjective arousal co-occurs with a specific pattern of physiological changes (HR, skin conductance, activity of the m. zygomaticus), thus supporting romantic love being an unique intense positive emotion which overlaps, but can be distinguished from other positive emotions (67, 69). Correspondingly, romantic love is associated with a certain brain activation pattern which is distinct from other positive emotional states (66). This unique pattern of intense positive emotion and the concomitant activation of approach motivation elicited by partner pictures is reflected by differences in SCL and corrugator EMG between partner and happy pictures, and can account for the stronger effect of partners rather than happy strangers on pain modulation. 

Results with regard to HR were more complex. We did not observe a modulating influence of the pictures on pain-related changes in HR when focusing on the average HR during PVT1 and PVT2. Yet, when the difference in HR between the first and last 5 s during PVTs was considered, partner faces were associated with greater HR deceleration. Sustained HR deceleration reflecting pain-relief has been observed in previous studies (19, 37), thus masking a possible HR acceleration which has been demonstrated in response to the presentation of faces of loved ones without concomitant pain stimulation (32, 72). Due to the presentaion of three pictures in a row during each PVT, it was not possible to disentangle overlapping time courses of HR changes in response to the emotional cues as has been demonstrated in pure picture viewing paradigms (58). Clearly, our findings require replication.

Further, as suggested by the reviewer, we removed the explanation in lines 517-518 to avoid the ‘speculative feel’. Instead, we now specifically highlight which results can be explained by the ECON model and which ones cannot, and we propose how the theory of social signal value of emotion expression can help to explain the data we obtained. Specifically, the section now reads: 

With regard to strangers’ faces, happy, but not angry faces had a pain modulating effect as shown by pain self-report and corrugator activity. Few studies have investigated pain modulation using positive (e.g. happy) as well as negative facial expressions (e.g., angry, sad, fearful). Sad (58–60) or angry (21) faces have been associated with increased pain intensity, whereas viewing happy faces reduced pain intensity (58-60), except in one study reporting enhanced pain (21). From the perspective of the Emotional Control of Nociception model [ECON, (59)], the rather consistent finding of pain attenuating effect of happy faces, as in the present study, is only partially accounted for. Our finding is consistent with the assumption of the ECON model that pictures’ valence determines the direction of pain modulation. It is, however, more difficult to reconcile with the assumption that the modulating influence of valence depends on the level of arousal. Emotional faces compared to IAPS pictures are typically perceived as less arousing (22, 60). In the present study, partner, happy and angry faces were all rated as moderately arousing. The theory of social signal value of emotion expression (61) could offer an explanation for the finding that happy (and partner) faces had a pain modulating effect despite being moderately arousing. One of its basic assumptions is that an expressed emotion provides information not only about the expressor, but also about the situation, and vice versa, and that context information can influence the perceived emotion. From this perspective, happy (and partner faces) might be a safety cue and of particular importance in a threatening pain context (1, 23, 62, 63). This corresponds well with the finding that brain regions related to safety-signals are only activated during pain relief when partner pictures are presented (15). As postulated by Krahé and colleagues (1), when experiencing pain, one’s social interactions may serve as predictive signals of contextual threat or safety, thus modulating the salience of pain. In the PVT paradigm used here, partners may function as particularly powerful safety signals as also suggested by the obtained SCL increase.

6) The manuscript is generally written in an understandable and clear manner, yet it would benefit from review by a native English speaker. 

Thank you for pointing this out! We have carefully checked the paper for proper English writing style.

7) Line 446: Table 1 should be changed by Table 3. 

We are not completely sure that we have fully understood the suggestion made by the reviewer. Table 1 includes sample characteristics, i.e. demographic data, questionnaire data and data on the obtained correlations between the relationship and psychological factors with partner pain modulation, picture valence and arousal. We have moved Table 1 at the end of section 2, such that the reader is informed about the questionnaires that were used. Table 1 is correctly referred to in section 1 (“Participants”), and later in the result section (4. Correlations between partner-PM, partner-related and psychological variables). In order to make it easier for the reader, we felt that it is useful to summarize the descriptive data and the correlation coefficients in one table. 

Comments Reviewer #2: 

The main limitation of this study is the reduced generalizability, because the sample is small (less than 30 subjects) and consists of young healthy women (mean=23.34, SD=2.89) in a close long-term relationship (69.9% had been in the partnership longer than two years). 

While we concur with the reviewer that the reduced generalizability of the study findings is a limitation of the study, we would like to point out that the study sample is comparable to the directly pertinent previous studies both with regard to relying on a female sample as well as sample size. We now address the issue of sample selection and size in more detail in the limitation section in the Discussion. The section now reads:

The main limitation of our study is the reduced generalizability as our sample consisted of a comparatively small sample of young, healthy women in a close long-term relationship. Several reasons led us to focus on a female sample with the aim of controlling for various potential confounding factors. Gender effects have been documented when using facial expression as stimuli, for observers of facial expressions, and for experimenters (74). Moreover, male facial expressions of anger modulate the startle response more strongly than female facial expressions, possibly interacting with the gender of the participant (75). Also, there is evidence that women are better at judging emotions from facial expressions (76). In experimental settings, male participants report less pain and tolerate more pain in the presence of female experimenters, thus pointing to complex interactions between the gender of the participant, the gender of social cues and the experimenter (77, 78). To avoid these confounding effects, and to ensure better comparability with previous studies also relying on female participants (15, 16), the sample included only female participants. The sample was representative with regard to relationship quality. Specifically, relationship satisfaction was comparable to different non-clinical samples of married and dating couples (40, 70, 71). Similarly, self-reported stress-related partner support was well within the normal range of support levels to be expected in this age group (44). Even though the sample size might be considered somewhat limited, it is comparable or even larger than in previous experimental studies investigating the influence of social cues on pain (15–17). The within-subject and randomized design is a strength of the study as well as the z-score standardization to adjust for interindividual differences in psychophysiological responding.

Introduction, Page 4, line 70. Add in parentheses the acronym that is later used (line 

73).

As recommended, the acronym has been added. The sentence now reads:

When interpreting their results, authors referred mainly to the pictures’ affective value reasoning that partner pictures are perceived to be pleasant which in turn leads to a positive affect. Consistent with Motivational Priming Theory (MPT; 18), partner pictures attenuated the experienced pain.

Methods, Page 11, line 248. “The pain intensity ratings for VAS t1-t3 were averaged”. Did the authors check before averaging that there were no statistically significant differences?

We believe that our formulation might have been misleading. As is typical in experimental pain research entailing several trials, we averaged, for each participant, the pain ratings separately for each measurement t1-t3 over the trials per picture category, in order to get a more robust estimate of perceived pain at these measurement times. 

To clarify this, we amended the relevant sentence which now reads:

Ratings. For each picture category and each rating time (VAS t1, VAS t2, VAS t3), the mean pain intensity was calculated as the average over the three trials. 

Methods, Page 11, lines 249-251. “Partner pain modulation (partner-PM) was defined as the difference in pain intensity between neutral and partner faces (neutral-partner) for VAS t3”. Why the authors did not use the average?

We thank the reviewer for asking this question. We first would like to point out that this difference score was only used for the correlational analysis. Given that we did not expect a modulatory effect of the neutral faces, and given the variation in pain intensity ratings across participants, this difference score allowed to determine the relative modulation by partner faces for each participant individually regardless of how they rated their pain in general. To convey this more succinctly, we changed the wording accordingly. It now reads:

For the correlational analyses, we defined partner pain modulation (partner-PM) as the difference in pain intensity between trials with neutral and partner faces (neutral – partner) at VAS t3, thus allowing to account for individual differences in pain experiences.

Methods, Pages 12-13, lines 259-287. The authors should clarify which analysis measure they used for each psychophysiological variable. In the case of SCL and corrugator EMG, did the authors measure mean amplitude?

Again, we apologize for not having described the recording and preprocessing of the psycho-physiological data in sufficient detail. We carefully checked sections “3.3.2. Psychophphysiological data” and “5.1. Data preprocessing” such that, for each psychophysiological measures, these details are provided. To illustrate this, the following information was added:

The mean SCL (mS) for 500 ms epochs was calculated. [SCL]

Mean EMG activity (µV) was calculated for 500 ms epochs. [corrugator EMG]

Methods, Page 14, line 309. It should be added that all participants also evaluated the valence and arousal of the stimuli.

Thanks for noting this. Please see our answer to Reviewer 1’s comment #3, particularly 3d. 

Methods, Page 14, lines 314-317. Again, it is not very clear to me how the authors have analyzed the physiological data. They have performed averages per trial (groups of 6 images of a single category), but the presentation of each image must generate a physiological response (which, in the case of SCL, is very slow).

We apologize that the description of how the psychophysiological data were analyzed was not sufficiently clear. The method section is revised and more details are now provided. There are also separate sections for data preprocessing (5.1.) and statistical analysis (5.2.). In addition, it is now more clearly described that the psychophysiological data were first segmented into 500 ms epochs and then averaged for PVT1 and 2, during which pictures of the same category were presented, thus inducing a somewhat prolonged affective state which we were primarily interested in. The relevant sentences read:

Preprocessing and aggregation of the psychophysiological data was done with customized Matlab programs. To reduce inter-individual differences, all physiological raw data were individually z-transformed [zi = (xi-Mi)/SDi]. As we used tonic heat pain stimulation, we were interested in psychophysiological responses during the heat pain stimulation. Therefore, after standardization (z-scores), we computed mean values for seven 5 s intervals per trial (BL, pic1-pic6) which were further collapsed over the three trials per picture category. Finally, averages for PVT1 (i.e. mean of pic1 to pic3) and for PVT2 (i.e. mean of pic4 to pic6) were computed, yielding mean values for PVT1, PVT2, and BL (see Figure 1b).

Methods, Page 14, line 318. Add the SPSS version used.

The SPSS version number has been added: 

All statistical analyses were calculated using IBM SPSS Statistics (version IBM 20.0.0, Armonk, NY) and p-values were set to .05.

Results, Page 16, lines 370-371. If during the individual adjustment of the intensity of the painful stimulus, the VAS had to be above 10 points (50%), why is the range so wide (2.13-16.20) in the experimental condition?

We thank the reviewer for bringing this issue to our attention. Indeed, we used a very elaborate individual adjustment procedure to determine each participant’s experimental temperature in which we defined several criteria to ensure heat stimuli were perceived as sufficiently painful. When re-reading the manuscript we noticed that reporting the mean pain intensity ratings for VASt1 across all trials is confusing and not meaningful because it reflects changes in pain intensity over time and upon repeated painful stimulation and, thus, is not the result of the adjustment procedure. Most crucially, we would like to point out that the difference in pain intensity between t1 and t2/t3 was analyzed per picture category and not across all pictures. Hence, VASt1 always refers to the pain intensity just before viewing the pictures separately in each category. As illustrated in Figure 2, there were no differences in baseline pain intensity (VASt1) between picture categories. We now report the average pain intensity rating at baseline (VASt1, i.e. before any picture viewing) in the very first experimental trial which is the first pain intensity rating after the adjustment procedure. The sentence reads now:

The mean pain intensity rating at t1 (i.e. before any picture viewing) in the first experimental trial was M = 11.1 (SD = 3.98).

Results, Page 17, lines 376-377. It is evident that no significant differences will be found in the pain intensity ratings at t1, because it is before the presentation of the images (or the BL of each trial). I suggest it be written as on line 399.

Thank you for pointing this out. The sentence now reads:

At baseline, there were no significant differences in pain intensity ratings (VAS t1) between the five different picture categories (Ms = 7.68 – 8.24, ps > .37, ds < 0.17).

Results, Page 18, line 401. “The effect was less pronounced at PVT2”. Did the authors find statistically significant differences?

Thank you for noting this unclear formulation. In order to make it easier for the reader to follow, we now report first which time segment is referred to. We have re-worded the sentence, which now reads:

During PVT 1, viewing partner faces elicited the highest SCL compared to all other picture categories (PVT1: ds = 1.01 – 1.59). Similarly, during PVT2, partner faces were associated with higher SCL compared to objects (d = 1.16) and neutral faces (d = 1.77) (see Figure 4a for contrasts). Further, during PVT2, SCL was larger for happy compared to neutral faces (d = 0.89). Finally, during PVT2, SCL was higher when viewing partner compared to happy faces (d = 0.82), and when viewing angry compared to neutral faces (d = 0.84), though not reaching statistical significance (p = .091 partner vs. happy; p = .056 angry vs. neutral).

Furthermore, at the beginning of the subsections in the result section, we now refer to the figure which illustrates the descriptive statistics and p-values for each physiological response. 

Results, Page 18, lines 414-421. HR deceleration analysis should be described in the methods section. Why did the authors not do this type of analysis with the other two physiological measures?

As suggested by the reviewer, we now describe in more detail how the HR data were analyzed, and especially how HR change from early to late PVTs was calculated. For better illustration, we have also added figures for all three physiological measures showing the changes over time (Figure 3 A-C). Unlike for SCL/EMG, HR changes in response to stimuli are usually characterized by phasic patterns of acceleration/¬deceleration which are interpreted as having a different functional meaning, e.g. orienting or defensive response (28, 58). Due to the consecutive presentations of the pictures, we were unfortunately not able to fully disentangle the overlapping time courses in HR change in response to the pictures. Hence, the HR deceleration analysis we performed can only provide first evidence of how picture category impacts on HR. We have addressed this issue in more detail in the discussion section. 

In addition, due to different time courses of HR changes during PVT1 and PVT2 (see Figure 3), difference scores between the first and last 5 s segments of PVT1 and PVT2 were computed, thus HR difference scores above 0 indicate HR deceleration between the early and late phase of picture viewing.

Results, Page 20, lines 453-456. The authors should describe in the methods section how controlled for arousal and helplessness in correlational analyses.

We apologize for not having described this more clearly. In order to identify the unique contribution of a variable to partner PM, we calculated partial correlations whenever a variable was significantly associated both with partner-PM and another of the variables considered in the correlation analyses. Upon your recommendation (see below), we added this information in the methods section. 

Discussion, Page 23, line 541. If the authors have performed partial correlations, they should describe in the methods section.

Thanks for bringing this to our attention. We have added a sentence in the methods section (5. Data analysis) to clarify this. We now also have labeled the reported partial rs more clearly. This sentence now reads:

In order to determine the unique relationship of a variable with partner-PM, partial correlations were calculated, whenever a variable correlated significantly both with partner-PM and another of the considered variables.

---

## [Decision Letter · Decision Letter 1]

15 Jun 2021

PONE-D-20-37976R1

Pain modulation by your partner: an experimental investigation from a social-affective perspective

PLOS ONE

Dear Dr. Kappesser,

i have now received comments from the two original reviewers. They were satisfied with your revision. However, reviewer 2still  made some minor recommendations regarding some typos and style. While i am tentatively accepting your manuscript, i would like that you consider these comments in a final version of your manuscript:

I consider that the authors have properly addressed all the reviewers’ comments, significantly improving the manuscript. However, the discussion section have some typing, spelling or grammatical errors (in the revised manuscript, line 509 (replace ":" by "."), line 530 (replace "extends" by "extend"), add "," after all "i.e." and "e.g.", replace "with regard" by "regarding", line 578 (replace "could offer an explanation" by "could explain"), lines 583-584 (replace "This corresponds well with..." by "This approach is consistent with..."), line 621 (replace "m. zygomaticus" by "zygomaticus muscle"), line 636 (replace "presentaion" by "presentation", line 646 (add "-" to "non clinical")) and is quite unwieldy. The authors should consider rewrite the eighth paragraph of the discussion section, because this issue had already been extensively addressed in previous paragraphs.

We look forward to receiving your revised manuscript.

Kind regards,

José A Hinojosa, Ph.D.

Academic Editor

PLOS ONE

Journal Requirements:

Reviewers' comments:

Reviewer's Responses to Questions

**Comments to the Author**

1. If the authors have adequately addressed your comments raised in a previous round of review and you feel that this manuscript is now acceptable for publication, you may indicate that here to bypass the “Comments to the Author” section, enter your conflict of interest statement in the “Confidential to Editor” section, and submit your "Accept" recommendation.

Reviewer #1: All comments have been addressed

Reviewer #2: All comments have been addressed

2. Is the manuscript technically sound, and do the data support the conclusions?

Reviewer #1: Yes

Reviewer #2: Yes

3. Has the statistical analysis been performed appropriately and rigorously? 

Reviewer #1: Yes

Reviewer #2: Yes

4. Have the authors made all data underlying the findings in their manuscript fully available?

Reviewer #1: Yes

Reviewer #2: Yes

5. Is the manuscript presented in an intelligible fashion and written in standard English?

Reviewer #1: No

Reviewer #2: No

6. Review Comments to the Author

Reviewer #1: I have read the authors’ responses to my comments and suggestions and I am quite happy with most of them. With the aim of not extending this review process beyond reason, I only would like to recognize the good job carried out by the authors in updating both the Method and Discussion sections.

Reviewer #2: (No Response)

7. PLOS authors have the option to publish the peer review history of their article (what does this mean?). If published, this will include your full peer review and any attached files.

Reviewer #1: No

Reviewer #2: No

---

## [Author Response · Author response to Decision Letter 1]

15 Jun 2021

Dear Professor Heber, 

Thank you for your letter and for passing on the comments of Reviewer #2 to which we reply step-by-step.

I consider that the authors have properly addressed all the reviewers’ comments, significantly improving the manuscript. However, the discussion section have some typing, spelling or grammatical errors (in the revised manuscript, line 509 (replace ":" by "."), line 530 (replace "extends" by "extend"), add "," after all "i.e." and "e.g.", replace "with regard" by "regarding", line 578 (replace "could offer an explanation" by "could explain"), lines 583-584 (replace "This corresponds well with..." by "This approach is consistent with..."), line 621 (replace "m. zygomaticus" by "zygomaticus muscle"), line 636 (replace "presentaion" by "presentation", line 646 (add "-" to "non clinical")) and is quite unwieldy. 

We corrected the typing, spelling and grammatical errors in lines 530, 578, 583/584, 621, 636, 646, added commas after all “i.e.” and “e.g.” and replaced “with regard to” by “regarding”.

The authors should consider rewrite the eighth paragraph of the discussion section, because this issue had already been extensively addressed in previous paragraphs.

We added the eighth paragraph particularly because of comment 5 by Reviewer #1 stating that “Several passages in the Discussion section should be much more detailed and elaborated. Whereas SCL effects for partners’ faces are clearly exposed, functional mechanisms underlying such results are not provided. This idea should be more developed, and an overview of the previous literature is required in order to interpret the functional meaning and implications of these results. The explanation included in lines 517-518 has a speculative feel with respect to data obtained in the present study.”

In order to comply with Reviewer #1’s suggestions, paragraph 8 in the revised version now not only addresses the results for the psychophysiological measures in much more detail (and extends the points that are discussed in other paragraphs in the Discussion), but also outlines possible underlying functional mechanisms such as romantic love that have not been addressed before. Therefore, we feel that paragraph 8 does not merely reiterate the content of previous paragraphs. Taking into consideration Reviewer # 1’ comment, we would like to leave paragraph 8 unchanged and hope that Reviewer #2 can follow our line of arguments.

---

## [Editor Report · Decision Letter 2]

21 Jun 2021

Pain modulation by your partner: an experimental investigation from a social-affective perspective

PONE-D-20-37976R2

Dear Dr. Kappesser,

We’re pleased to inform you that your manuscript has been judged scientifically suitable for publication and will be formally accepted for publication once it meets all outstanding technical requirements.

Kind regards,

José A Hinojosa, Ph.D.

Academic Editor

PLOS ONE
---

## [Editor Report · Acceptance letter]

13 Jul 2021

PONE-D-20-37976R2 

Pain modulation by your partner: an experimental investigation from a social-affective perspective 

Dear Dr. Hermann:

I'm pleased to inform you that your manuscript has been deemed suitable for publication in PLOS ONE. Congratulations! Your manuscript is now with our production department. 

Kind regards, 

on behalf of

Dr. José A Hinojosa 

Academic Editor

PLOS ONE